# INTENTGPT: FEW-SHOT INTENT DISCOVERY WITH LARGE LANGUAGE MODELS

## ABSTRACT

In today's digitally driven world, dialogue systems play a pivotal role in enhancing user interactions, from customer service to virtual assistants. In these dialogues, it is important to identify user's goals automatically to resolve their needs promptly. This has necessitated the integration of models that perform Intent Detection. However, users' intents are diverse and dynamic, making it challenging to maintain a fixed set of predefined intents. As a result, a more practical approach is to develop a model capable of identifying new intents as they emerge. We address the challenge of Intent Discovery, an area that has drawn significant attention in recent research efforts. Existing methods need to train on a substantial amount of data for correctly identifying new intents, demanding significant human effort. To overcome this, we introduce IntentGPT, a novel method that efficiently prompts Large Language Models (LLMs) such as GPT-4 to effectively discover new intents with minimal labeled data. IntentGPT comprises an *In-Context Prompt Generator*, which generates informative prompts for In-Context Learning, an *Intent Predictor* for classifying and discovering user intents behind utterances, and a *Semantic Few-Shot Sampler* which leverages embedding similarities for selecting the closest examples from the labeled data. Our experiments show that Intent-GPT outperforms previous methods that require extensive domain-specific data and fine-tuning, in popular benchmarks, including CLINC and BANKING.

## 1 INTRODUCTION

Intent Discovery is a natural language processing (NLP) task that involves classifying user-written sentences into intents within an *open-world* context, where new intents will emerge and need to be identified. This task is crucial for modern dialogue systems (Degand & Muller, 2020), allowing them to decipher user queries, whether they involve seeking information, making requests, or expressing opinions, and steering the conversation appropriately. This task is dynamic since user intents change over time and new digital tools continually emerge, expanding the range of intents. Intent prediction needs to robustly adapt to this open-world scenario to stay responsive to changing user needs.

Intent Detection has received considerable research in recent years (Lin et al., 2023; Costello et al., 2018; FitzGerald et al., 2022; Jayarao & Srivastava, 2018; Zhang et al., 2020). These works approach the problem from a closed-world setting of known classes, while often allowing for predictions on out-of-scope examples into an additional "other" class. Our focus lies in Intent Discovery (Shi et al., 2018; Zhang et al., 2022), which assumes an open-world setting where novel intents must be discovered. This approach aligns better with real-world applications and their inherent complexity.

The task of Intent Discovery has seen notable exploration through the application of clustering methods and semi-supervised training, as discussed in recent literature (Kumar et al., 2022; Zhang et al., 2021b). However, it is essential to recognize that the effectiveness of these approaches often hinges on the availability of substantial labeled data and multi-stage training procedures. Implementing such methods and refining their performance through extensive training and validation can be a resource-intensive endeavor. In many cases, the process may involve fine-tuning models on domain-specific data, which requires careful curation and labeling efforts.

Given these challenges, using pre-trained large language models (LLM) emerges as a compelling alternative. Pre-trained models, such as GPT 3-4 (Brown et al., 2020; OpenAI, 2023), have undergone

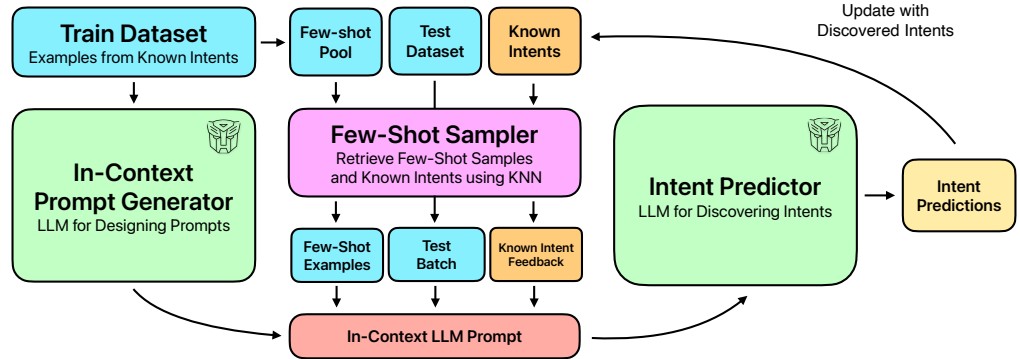

Figure 1: **IntentGPT architecture.** The system consists of two frozen Transformer LLMs. The In-Context Prompt Generator receives samples from the training dataset and is devoted to generating a prompt using domain context. The generated prompt is filled with in-context data gathered by the Few-Shot Sampler, containing few-shot examples and known intents (Known Intent Feedback). Finally, the Intent Predictor will use the generated prompt to perform the Intent Discovery task.

extensive training on vast and diverse text corpora encompassing a wide range of domains and contexts. This pre-training endows them with a broad understanding of language, making them adept at handling complex and diverse user inputs (Bubeck et al., 2023). This offers a new perspective to handling open-set problems. As framed by Scheirer et al. (2012), open-set problems pertain to managing the empirical risk space connected with unknown classes. We can consider a model such as GPT-4 to possess extensive knowledge of the world due to its training data. This suggests that the model should not have a problem with the concept of empirical risk space in an open-world.

With the increasing availability of open-source LLMs[1], the community has embraced a new paradigm for adapting these models to downstream tasks by crafting carefully designed prompts (Liu et al., 2023). This paradigm shift has demonstrated that, with meticulous prompt construction, these models can deliver precise predictions. Notably, prompt design has proven remarkably influential in shaping LLM behavior Brown et al. (2020). Prompts typically comprise a task description and several canonical examples, constituting a framework for in-context learning (ICL) (Xie et al., 2021).

In this work, we propose **IntentGPT**, a method for Few-Shot Intent Discovery using pre-trained LLMs. The architecture is presented in Figure 1. Our design offers a general solution that requires no training and relies strictly on model generalization and ICL capabilities. Our method assumes a strong world knowledge of large pre-trained models like GPT-3.5, GPT-4, and Llama-2 (Touvron et al., 2023), and leverages them for 1) designing In-Context Prompts and 2) performing Intent Discovery. As shown in Figure 1, we propose techniques to promote intent reusability and inject semantically meaningful examples to the ICL prompt, describing the context of the data domain.

**Contributions.** In summary, our contributions are as follows: **i)** Introduce and open-source IntentGPT, a framework for Intent Discovery that uses pre-trained LLMs for Few-Shot In-Context Learning. **ii)** Show that LLMs can now perform tasks that previously required extensive data and training, by using innovative techniques for prompt design and semantic few-shot selection. **iii)** Conduct experiments on challenging Intent Discovery benchmarks, showcasing our method's competitive performance against models demanding more data and training. **iv)** Provide a comprehensive analysis of hyperparameter influence on metric performance in the novel few-shot ICL setting using frozen LLMs.

The paper is structured as follows. Section 2 presents relevant works in the context of intent discovery and in-context learning. Section 3 presents the proposed method, IntentGPT. Section 4 describes the experimentation setup, datasets used, and evaluation framework. Section 5 shows relevant results of our method. Finally, Section 6 presents conclusions.

---

[1] https://huggingface.co/spaces/HuggingFaceH4/open_llm_leaderboard

## 2 RELATED WORK

To discover new intents, we build from the framework of open set recognition (Scheirer et al., 2012). Open set recognition has been widely studied (Geng et al., 2020) in the domain of computer vision to handle the concept of an open-world. Recent work (Zhang et al., 2021b; Kumar et al., 2022) has shown interest in the same underlying problem with natural language processing and applies them to the task of Intent Detection and Discovery.

**Intent Detection.** Intent Detection focuses on classifying pre-defined user intents, typically through text sentences (Liu et al., 2019). Bunk et al. (2020) introduce a transformer-based (Vaswani et al., 2017) architecture that jointly learns entity recognition and Intent Detection. Lin et al. (2023) detects the lack of annotated pairs of utterances and intents as the primary challenge for Intent Detection, and leverages in-context learning to extract high-quality data samples. Researchers have also explored contrastive methods aimed at learning discriminative features from utterances (Zhang et al., 2021c).

Despite the attention and promising results in this area, this scenario lacks realism as it does not account for the emergence of new, out-of-distribution (OOD) intents in real-world settings.

**Open-World Intent Detection.** aims to classify known intents and identify OOD examples into an unknown class following an open-world perspective. Previous work by Shu et al. (2017) was one of the first to approach text classification from an open-world perspective. Their method used a CNN with a final 1-vs-rest layer and a sigmoid activation function. Recent methods to approach this problem have used KNN-contrastive learning (Zhou et al., 2022), synthetic examples Zhan et al. (2021), and adaptive decision boundaries (Zhang et al., 2021a). Other techniques seek to leverage traditional fine-tuning (Devlin et al., 2018) paradigms but threshold the softmax outputs (Zhang et al., 2020) or utilize a manually crafted out-of-distribution dataset (Zhang et al., 2021d).

While detecting OOD samples is a significant step toward a real-world scenario, it alone does not fully address the issue. Ideally, we want our model to proactively generate new intents by design.

**Intent Discovery.** In the context of Intent Discovery, the model operates under the assumption of an *open-world*, where certain intents are known while others remain undiscovered, continuously advancing in its quest to uncover them. The task presents a challenge as the model must exhibit discriminative capabilities, not only for classifying known examples but also for effectively distinguishing samples associated with never-seen intents that may be present in the test set. Prior methods involved a two-stage process: first, training a text encoder to learn robust representations, and second, obtaining clusters for identifying new intents. Notably, Zhang et al. (2021b) presents DeepAligned, which employs BERT (Devlin et al., 2018) to extract intent embeddings and introduces an innovative alignment strategy for learning cluster representations. SCL, as proposed by Shen et al. (2021) adopts a Siamese network and a contrastive loss to discern whether two utterances align with the same intent. DSSCC (Kumar et al., 2022) uses labeled utterances linked to known intents for pretraining a SentenceBert model (Reimers & Gurevych, 2019). Subsequently, it employs contrastive learning on both labeled and unlabeled samples to refine and learn effective clusters. Recently, Zhou et al. (2023) proposed a probabilistic framework that employs expectation maximization treating the intent assignments as latent variables. IDAS (De Raedt et al., 2023) is the method that is closest to ours, being the first that uses LLMs on Intent Discovery. They pose the task as an abstractive summarization problem, where summaries are generated from utterances and further clustered.

While these methods have shown promising results, their reliance on multiple training stages and domain-specific data can be suboptimal. Our proposed approach offers an alternative paradigm by eliminating the need for training and optimizes data usage through a novel few-shot ICL setting.

**In-context Learning.** In-context learning (ICL) serves as a technique for customizing LLMs to execute downstream tasks without necessitating fine-tuning. ICL involves the conditioning of a pretrained LLM with a task description and optionally some problem demonstrations, such as injecting input-label pairs into the prompt (few-shot ICL) (Brown et al., 2020; Xie et al., 2021). The application of few-shot ICL has yielded substantial enhancements in metric performance for LLMs across a spectrum of NLP tasks (Liang et al., 2022). A major driver of this shift was the rise of emergent properties of LLMs from scaling them up (Wei et al., 2022). Bubeck et al. (2023) study the rising capabilities and implications of largely scaled language models like GPT-4, and showcase their performance on complex downstream tasks like coding, vision, or medicine among others.

While many ICL setups utilize only a few labeled examples, ongoing research explores prompt optimization. In this regard, two key challenges emerge: selecting effective prompts and identifying suitable few-shot examples. To address prompt selection, Brown et al. (2020) propose *prompt engineering* involving iterative trial-and-error approaches, albeit with human effort, potential errors, and context length limitations. AutoPrompt (Shin et al., 2020) tackles this issue through a gradient-based search strategy. Prompt Tuning (Lester et al., 2021) has gained popularity by introducing learnable weights for specific tokens while maintaining frozen LLM weights. Addressing the challenge of identifying optimal few-shot samples, prior approaches have focused on example retrieval (Rubin et al., 2022). Recent work by Gao et al. (2023) highlights the effectiveness of combining ICL with semantic similarity methods, outperforming fine-tuning in various downstream tasks. Wang et al. (2023) showcased GPT-4's exceptional capabilities in domains like Minecraft, using ICL for code generation and curriculum development, suggesting potential applications in an *open-world*.

Our approach for prompt design involves using strong LLMs, such as GPT-4, prompted with labeled examples to generate a precise prompt that offers domain-specific guidelines and context. For selecting few-shot examples, we employ a retrieval module to extract semantically relevant samples.

# 3 METHOD

This section introduces IntentGPT, a Few-Shot In-Context Learning approach tailored for Intent Discovery. As shown in Figure 1, IntentGPT comprises two pre-trained transformer LLMs. The *In-Context Prompt Generator* LLM autonomously designs a prompt for the Intent Discovery task, leveraging pairs of utterances and intents from the training set to grasp the domain and contextual information. This language model is prompted to assist in generating a comprehensive prompt. On the other hand, the *Intent Predictor* LLM utilizes the generated prompt, with injected few-shot samples, test examples to perform inference, and optionally, a list of known intents to keep track of the discovered intents and reuse them if needed (*Known Intent Feedback*) to perform intent predictions. These two LLMs can either be the same model with distinct contexts and tasks or entirely different models. Furthermore, the intermediate *Few-Shot Sampling* module, positioned in between, implements various prompting techniques to infuse meaningful information into the prompt. Figure 2 depicts the two stages of our method with real examples from CLINC.

The following sections describe the different modules and prompt features available in IntentGPT.

## 3.1 IN-CONTEXT PROMPT GENERATOR

As ICL performance is substantially influenced by prompt quality, our focus centers on establishing a systematic methodology for crafting effective prompts specifically tailored for the Intent Discovery task. We focus on distinct and diverse domains, such as finance, travel, or daily needs. To achieve this goal, we propose the utilization of a robust LLM, such as GPT-4, for prompt design. Specifically, we use examples from the training dataset to give context, and ask the In-Context Prompt Generator (ICPG) to solve the task of generating a prompt for Intent Discovery. We show that this improves the quality of the predictions through ablation studies.

We initiate the prompt design process by first engaging with the LLM (GPT-4 in our experiments) to ensure the correct output format. Subsequently, we choose a random selection of $x$ examples per known intent (e.g., $x = 2$ in our experiments) and integrate them into the prompt. This step is essential to provide sufficient context regarding the task and the data domain. By conditioning the LLM on a substantial amount of data, we enable it to grasp the intricacies of both the data and context, thereby facilitating the design of a prompt for another LLM tasked with solving the Intent Discovery task. In the ICPG prompt, we emphasize that the prompt must be concise and explicitly instruct that the 'unknown' intent should never be assigned. For the complete prompt at this stage, please refer to Appendix A.2 (Prompt A). This process is a one-time operation for a given number of known intents $n$ and the desired number of examples per intent $x$. Once generated, the prompt is retained for future use within the same setting. For examples of prompts generated by the ICPG, please refer to Appendix A.2 (prompts B and C).

**In-Context Prompt Generation**

**Intent Predictor**

Figure 2: Pipeline of IntentGPT with real examples. The top part displays the automatic prompt generation process. The bottom part shows the inference process, where we use the generated prompt and few-shot examples for discovering intents. We do not show the Few Shot Sampler for simplicity.

## 3.2 FEW-SHOT SAMPLER

We introduce a module designed to extract training examples and incorporate them into the ICL prompt. This module has access to a Few-Shot Pool containing available examples and employs techniques to select meaningful few-shot samples, providing valuable information to the LLM. This sampler can just be random, but in practice, we employ a Semantic Few-Shot Sampling technique, which finds samples based on embedding similarity with the test batch.

**Few-Shot Pool.** The Few-Shot Pool comprises a subset of the training dataset, containing 10% of the samples for each known intent, for all known intents $n$. This set of samples is utilized to retrieve few-shot examples during inference.

**Semantic Few-Shot Sampling (SFS).** We draw from the work of Liu et al. (2021) and Retrieval Augemted Generation (RAG) Lewis et al. (2020); Izacard et al. (2022) to perform a K-Nearest Neighbors (KNN) Semantic Sampling. In our work, we use, SentenceBert (Reimers & Gurevych, 2019) (SBERT) to embed the sentences and employ cosine distance as our similarity measure to determine those examples close in the latent space. The selected examples are then concatenated together to become our sequence of in-context examples that are fed to the model.

## 3.3 KNOWN INTENT FEEDBACK

During inference with the test set, the system will discover novel intents. Accessing both known and discovered intents is crucial for making accurate predictions. To address this need, we propose using *Known Intent Feedback* (KIF), a straightforward technique that conditions the LLM to reuse intents from the database by injecting the known and discovered intents directly into the prompt. Being $n$ the number of known intents, and $m$ the number of discovered intents at the end of an iteration, the system updates the database of intents to a size $n + m$. This operation makes the model aware of the accessible intents and allows it to reuse them. This process is shown in Figure 1 and 2. See Appendix A.2 for the specific prompt we use.

**Semantic Known Intent Feedback Sampling (SKIF).** Following the same technique described in SFS, we incorporate the option to reduce the list of known intents included in the prompt using KNN. This introduces a variable, denoted as $n_{\text{SKIF}}$, which determines the number of intents injected into the prompt to inform the model about the current known and discovered intents. This process serves two main purposes. First, it aims to introduce ICL examples that are semantically similar to the current test batch. Second, it optimizes the use of the language model's context length by avoiding the injection of the entire list of known intents into the prompt. This approach allows us to

strike a balance between metric performance and query efficiency. If this feature is not activated, all known intents pass through directly to the prompt.

## 3.4 INTENT PREDICTOR

The Intent Predictor LLM is conditioned with the generated In-Context Prompt encompassing a task description, few-shot examples, and optionally, the unique set of known and discovered intents. Test examples are introduced at the end of the prompt to facilitate inference. This prompt is constructed using a template that precisely defines its content and format (see prompt at Appendix A.2). To ensure accurate predictions, we include explicit instructions in the prompt, specifying that the model should solely focus on predicting test labels and avoid extracting few-shot samples. We also define the desired output format to simplify parsing. After obtaining predictions, we check for newly discovered intents and incorporate them into the *Known Intents* database, expanding the list of intents considered by the model moving forward.

**Clustering of Intents.** We adopt the standard choice (Zhang et al., 2021b) of using K-Means for clustering intents. We compute SBERT embeddings on predicted and ground truth intents and perform K-Means clustering. We automatically compute the $K$ in K-Means by first using DB-SCAN (Ester et al., 1996) with $\epsilon = 0.5$ on the predicted embeddings.

## 4 EXPERIMENTAL SETTING

In this section, we describe the datasets, the evaluation framework, and the Intent Discovery metrics.

**Datasets.** We evaluate IntentGPT on two datasets, namely CLINC (Larson et al., 2019) and BANK-ING (Casanueva et al., 2020). CLINC comprises pairs of text utterances and intents from user interactions across 10 different domains, including work, travel, and banking, totaling 150 unique intents. The dataset encompasses 18,000 training examples, 2,250 validation examples, and 2,250 test examples. BANKING follows a similar format of utterances and intents but focuses exclusively on the finance domain, encompassing 77 distinct intents. This dataset contains 9,003 training examples, 1,000 validation examples, and 3,080 test examples.

**Evaluation setting and metrics.** We adopt the evaluation framework proposed by Zhang et al. (2021b), which includes defining a Known Intent Ratio (KIR) and allocating a percentage of training samples per each known intent, typically set at 10% i.e. our Few-Shot Pool. In our Few-Shot ICL approach, we use $n_{shots}$ number of examples in the prompt, which is a much lower-data setup compared to previous works. In the context of Intent Discovery, evaluation metrics pivot towards the utilization of clustering-based measures, which assess the models' effectiveness in assigning utterances to their corresponding intents. To evaluate clustering results we adopt Normalized Mutual Information (NMI), Clustering Accuracy (ACC), and Adjusted Rand Index (ARI) metrics as proposed in the literature (Kumar et al., 2022). These metrics are widely recognized in the field as effective measures for assessing clustering quality. These metrics quantify the degree of alignment between predictions and the ground truth following the clustering of intent representations, i.e., text embeddings Lin et al. (2020); Zhang et al. (2021b). Additionally, we introduce the Number of Discovered Intents (NDI) as a metric to assess how the model is performing in terms of discovering intents and compare it to the ground truth..

## 5 RESULTS AND DISCUSSION

We present the results of IntentGPT when powered by three different pre-trained LLMs, namely GPT-3.5, GPT-4, and Llama-2. See Appendix A.3 for details on these LLMs. We compare our method against the baselines on the CLINC and BANKING test sets. During inference, we use a batch size of 16, which offers a good trade-off between context length and inference cost.

**Effectiveness of IntentGPT** The main result of our paper is presented here. We evaluate the performance of IntentGPT models on the KIR = 0.75 setting, a commonly used configuration for Intent Discovery evaluation. Results are presented in Table 1. We report the performance of our method in the 0-shot, 10-shot, and 50-shot settings. For the 0-shot setting, we do not use any few-shot samples,

| Setting | Method | CLINC | | | BANKING | | |
|---|---|---|---|---|---|---|---|
| | | NMI ↑ | ARI↑ | ACC↑ | NMI ↑ | ARI↑ | ACC↑ |
| Unsupervised | KM (MacQueen et al., 1967) | 70.89 | 26.86 | 45.06 | 54.57 | 12.18 | 29.55 |
| | AG (Gowda & Krishna, 1978) | 73.07 | 27.70 | 44.03 | 57.07 | 13.31 | 31.58 |
| | SAE-KM Xie et al. (2016) | 73.13 | 29.95 | 46.75 | 63.79 | 22.85 | 38.92 |
| | DEC Xie et al. (2016) | 74.83 | 27.46 | 46.89 | 67.78 | 27.21 | 41.29 |
| | DCN (Yang et al., 2017) | 75.66 | 31.15 | 49.29 | 67.54 | 26.81 | 41.99 |
| | DAC (Chang et al., 2017) | 78.40 | 40.49 | 55.94 | 47.35 | 14.24 | 27.41 |
| | DeepCluster (Caron et al., 2018) | 65.58 | 19.11 | 35.70 | 41.77 | 8.95 | 20.69 |
| | IDAS (De Raedt et al., 2023) | - | - | - | 80.43 | 53.31 | 63.77 |
| | IntentGPT-Llama-2$_{0\,shot}$ (ours) | 85.63 | 36.64 | 58.04 | 76.32 | 35.95 | 44.82 |
| | IntentGPT-3.5$_{0\,shot}$ (ours) | 90.04 | 65.62 | 73.68 | 76.58 | 40.30 | 54.82 |
| | IntentGPT-4$_{0\,shot}$ (ours) | 94.35 | 78.69 | 83.20 | 81.42 | 55.18 | 64.22 |
| Semi-Supervised | PCK-means (Basu et al., 2004) | 68.70 | 35.40 | 54.61 | 48.22 | 16.24 | 32.66 |
| | BERT-KCL (Hsu et al., 2017) | 86.82 | 58.79 | 68.86 | 75.21 | 46.72 | 60.15 |
| | BERT-MCL (Hsu et al., 2019) | 87.72 | 59.92 | 69.66 | 75.68 | 47.43 | 61.14 |
| | CDAC+ (Lin et al., 2020) | 86.65 | 54.33 | 69.89 | 72.25 | 40.97 | 53.83 |
| | BERT-DTC (Han et al., 2019) | 90.54 | 65.02 | 74.15 | 76.55 | 44.70 | 56.51 |
| | DeepAligned (Zhang et al., 2021b) | 93.89 | 79.75 | 86.49 | 79.56 | 53.64 | 64.90 |
| | DSSCC (Kumar et al., 2022) | 87.91 | 81.09 | 87.91 | 81.24 | 58.09 | 69.82 |
| | SCL (Shen et al., 2021) | 94.75 | 81.64 | 86.91 | 85.04 | 65.43 | 76.55 |
| | IDAS (De Raedt et al., 2023) | 93.82 | 79.02 | 85.48 | - | - | - |
| | LatentEM (Zhou et al., 2023) | 95.01 | 83.00 | **88.99** | 84.02 | 62.92 | 74.03 |
| Few-Shot ICL | IntentGPT-Llama-2$_{10\,shot}$ (ours) | 87.86 | 60.35 | 71.75 | 77.39 | 45.52 | 55.97 |
| | IntentGPT-Llama-2$_{50\,shot}$ (ours) | 91.71 | 68.10 | 77.95 | 81.56 | 56.68 | 69.83 |
| | IntentGPT-3.5$_{10\,shot}$ (ours) | 91.72 | 71.69 | 77.56 | 78.21 | 48.20 | 63.65 |
| | IntentGPT-3.5$_{50\,shot}$ (ours) | 95.60 | 76.97 | 84.86 | 81.79 | 56.60 | 68.77 |
| | IntentGPT-4$_{10\,shot}$ (ours) | 94.85 | 80.36 | 85.07 | 83.11 | 59.07 | 70.42 |
| | IntentGPT-4$_{50\,shot}$ (ours) | **96.06** | **84.76** | 88.76 | **85.94** | **66.66** | **77.21** |

Table 1: Comparison against unsupervised and semi-supervised models. The best models are shown in **bold**, and the second best are underlined. Baseline results are extracted from Kumar et al. (2022).

the prompt is simply a basic description of the task, and we activate Known Intent Feedback (KIF) to reuse intents. In the 10 and 50-shot settings, we utilize Semantic Few-Shot Sampling (SFS) and KIF at each iteration. Notably, IntentGPT-4 in the 50-shot setting outperforms prior methods in both unsupervised and semi-supervised scenarios, except for ACC on CLINC where LatentEM achieves a slightly higher score. IntentGPT-3.5 shows strong results competing with semi-supervised methods, both using 10 and 50-shot. Even in the 0-shot setting, which is considered unsupervised, our models outperform all previous methods and demonstrate overall competitive performance. One could argue that IntentGPT is so effective due to the underlying model being possibly trained in the datasets, we discuss this matter in Appendix A.6.

**Impact of Known Intent Ratio (KIR)** Here we show how IntentGPT performance changes with variations in the initial number of known intents (KIR). We compare few-shot and semi-supervised methods, on three settings KIR $\in \{0.25, 0.50, 0.75\}$. Results are presented in Figures 3 and 4. IntentGPT outperforms all previous methods in all KIR settings. We can observe a clear performance scale when comparing LLMs. GPT-4 is the winner with higher scores across settings, followed by GPT-3.5 and Llama-2. This is consistent with results seen in other popular NLP benchmarks[2]. Due to the scale in network parameters, amount of train data, and specific fine-tuning techniques, GPT-4 shows exceptional generalization behavior on Intent Discovery.

**Ablation on Prompt Features** We investigate the influence of the proposed prompt features on the performance of IntentGPT using GPT-3.5 and GPT-4 models. We use the KIR $= 0.75$ setting. When activating Few-Shot (FS) we introduce 10 shots, and when applying Semantic Known Intent Feedback (SKIF), we inject the 50 known intents that exhibit the highest similarity to the test utterances. Table 2 presents the results for the most significant combinations of prompt features, providing insights into their influence. Notably, the inclusion of Known Intent Feedback (KIF) yields

---

[2]https://llm-leaderboard.streamlit.app/

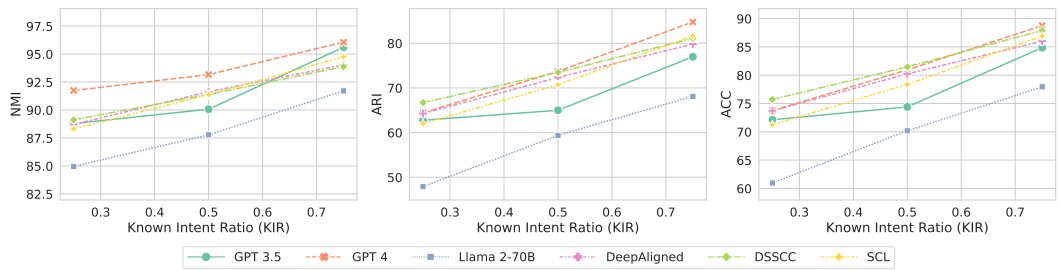

Figure 3: Results on CLINC of IntentGPT models compared with previous baselines, as a function of the Known Intent Ratio (KIR). Ours are GPT-3.5, GPT-4 and Llama-2 70B.

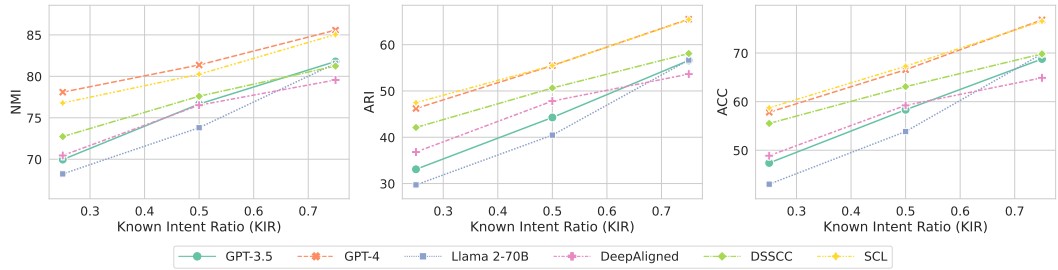

Figure 4: Results on BANKING of IntentGPT models compared with previous baselines, as a function of the Known Intent Ratio (KIR). Ours are GPT-3.5, GPT-4 and Llama-2 70B

substantial gains, as its absence results in a significant increase in the number of discovered intents, hence failing in other metrics. This is attributed to the model not being aware of the known intents and its inability to reuse them effectively. The utilization of few-shot learning (FS) demonstrates its advantages, particularly when applying Semantic Few-Shot Sampling (SFS), which consistently yields higher scores. It is worth noting that while SKIF does not result in significant gains. This outcome aligns with expectations, as the optimal scenario involves including all known intents in the prompt.

| Method | KIF | FS | SFS | ICP | SKIF | NMI ↑ | ARI ↑ | ACC ↑ | NDI ↓ | NMI ↑ | ARI ↑ | ACC ↑ | NDI ↓ |
|---|---|---|---|---|---|---|---|---|---|---|---|---|---|
| | **Prompt Features** | | | | | **CLINC** | | | | **BANKING** | | | |
| IntentGPT-3.5 | ✔ | | | | | 80.12 | 41.08 | 56.13 | 1484 | 68.26 | 21.70 | 31.74 | 997 |
| | ✔ | | | | | 90.26 | 65.53 | 74.42 | 241 | 76.58 | 40.30 | 54.82 | 120 |
| | ✔ | ✔ | | | | 90.42 | **66.53** | 75.30 | 176 | 77.17 | 45.59 | 59.32 | 115 |
| | ✔ | ✔ | ✔ | | | 90.71 | 65.29 | 76.60 | 159 | 78.79 | 48.65 | **64.89** | **83** |
| | ✔ | ✔ | | ✔ | | 92.31 | 65.26 | 78.04 | **149** | 77.90 | 45.02 | 58.35 | 130 |
| | ✔ | ✔ | ✔ | ✔ | | **93.07** | 65.39 | **78.95** | 147 | 80.05 | **50.12** | 63.49 | 152 |
| | ✔ | ✔ | ✔ | ✔ | ✔ | 91.77 | 65.62 | 75.78 | 204 | **80.54** | 43.44 | 51.06 | 574 |
| IntentGPT-4 | ✔ | | | | | 86.32 | 56.96 | 66.44 | 1038 | 72.00 | 30.94 | 38.12 | 1322 |
| | ✔ | | | | | 93.81 | 75.65 | 82.44 | **149** | 80.66 | 53.47 | 64.25 | 102 |
| | ✔ | ✔ | | | | 93.58 | 76.55 | 83.73 | 155 | 81.70 | 55.09 | 64.51 | 125 |
| | ✔ | ✔ | ✔ | | | 93.89 | 75.72 | 82.09 | 146 | 81.71 | 55.91 | 67.92 | **79** |
| | ✔ | ✔ | | ✔ | | 93.95 | 77.71 | 83.91 | 163 | 80.54 | 54.80 | 66.40 | 110 |
| | ✔ | ✔ | ✔ | ✔ | | **94.99** | **80.48** | **85.42** | 159 | **83.18** | **60.10** | **70.42** | 118 |
| | ✔ | ✔ | ✔ | ✔ | ✔ | 91.70 | 73.20 | 78.80 | 237 | 80.62 | 54.26 | 64.90 | 173 |

Table 2: Ablation study of the proposed prompt features. **KIF**: Known Intent Feedback, **SKIF**: Semantic Known Intent Feedback, **SFS**: Semantic Few-Shot, **ICP**: In-Context Prompt. For NDI, the best results are the closest to the ground truth (150 for CLINC and 77 for BANKING).

**Impact of increasing the number of shots** We conduct a set of experiments where we increase the number of few-shot examples injected in the prompt for ICL, to asses the gains we can obtain when showing more context examples to the LLM. We perform these experiments on all the KIR settings and compare them against two state-of-the-art baselines, namely DeepAligned and SCL which report results on the same settings. Results are depicted in Table 3. Our observations indicate that increasing the number of shots leads to significant performance improvements, particularly in

the KIR=0.75 setting. However, escalating the number of shots in the KIR = 0.25 scenario does not yield proportionate performance growth. This phenomenon is attributed to the model's limited number of absolute intents, leading to divergence over time.

| KIR | Method | Shots | CLINC | | | | BANKING | | | |
|---|---|---|---|---|---|---|---|---|---|---|
| | | | NMI ↑ | ARI↑ | ACC↑ | NDI↓ | NMI ↑ | ARI↑ | ACC↑ | NDI↓ |
| | DeepAligned | - | 88.71 | 64.27 | 73.71 | - | 68.88 | 35.49 | 47.58 | - |
| | SCL | - | 88.30 | 62.02 | 71.25 | - | 76.79 | **47.47** | 58.73 | - |
| 0.25 | | 10 | 88.03 | 55.29 | 66.93 | 136 | 76.25 | 42.39 | 55.91 | 99 |
| | | 20 | 91.13 | **66.57** | **74.18** | 169 | 76.81 | 43.95 | 55.58 | 90 |
| | IntentGPT-4 (ours) | 30 | **91.74** | 64.43 | 73.73 | 136 | 76.84 | 44.30 | 56.43 | 94 |
| | | 40 | 90.45 | 59.59 | 69.11 | **142** | 76.09 | 40.14 | 52.60 | **75** |
| | | 50 | 90.09 | 58.09 | 70.31 | 136 | **78.73** | 46.54 | **59.09** | 89 |
| | DeepAligned | - | 91.63 | 72.34 | 80.22 | - | 76.14 | 47.07 | 59.44 | - |
| | SCL | - | 91.38 | 70.71 | 78.36 | - | 80.25 | **55.50** | **67.28** | - |
| 0.5 | | 10 | 91.44 | 66.93 | 74.44 | 143 | **81.06** | 55.17 | 66.33 | 131 |
| | | 20 | 91.41 | 63.16 | 73.11 | 141 | 79.77 | 52.50 | 64.97 | 111 |
| | IntentGPT-4 (ours) | 30 | 92.72 | 72.47 | 79.16 | 141 | 80.43 | 53.21 | 66.79 | 102 |
| | | 40 | 92.93 | 71.96 | 77.47 | 140 | 79.71 | 51.26 | 63.02 | **88** |
| | | 50 | **93.16** | **73.69** | **80.94** | 155 | 79.45 | 50.22 | 62.76 | 92 |
| | DeepAligned | - | 94.03 | 79.82 | 86.01 | - | 78.77 | 52.11 | 63.68 | - |
| | SCL | - | 94.75 | 81.64 | 86.91 | - | 85.04 | 65.43 | 76.55 | - |
| 0.75 | | 10 | 94.69 | 79.56 | 85.07 | 164 | 83.46 | 59.91 | 70.39 | 109 |
| | | 20 | 95.22 | 81.51 | 86.22 | 163 | 84.49 | 63.73 | 74.03 | 114 |
| | IntentGPT-4 (ours) | 30 | 95.75 | 83.89 | 87.82 | 166 | **85.32** | 65.30 | 76.33 | 98 |
| | | 40 | 95.97 | 84.08 | 88.49 | **156** | 85.01 | 64.07 | 74.58 | 95 |
| | | 50 | **96.06** | **84.76** | **88.76** | 159 | 85.07 | **66.66** | **77.21** | **88** |

Table 3: Performance gains of IntentGPT when we increase the number of shots for few-shot ICL using GPT-4. Baseline results are extracted from Kumar et al. (2022) and Zhang et al. (2021b).

**Limitations** We outline the limitations of our model. We have a strong restriction on the context length of the LLM, and the context limit may vary over models. The attention operation makes computational and monetary costs increase as context increases. Much research is being held in this matter Bulatov et al. (2023); Dao et al. (2022), hence future models might not suffer from this issue. Running the model on large LLMs like GPT-4 can be expensive and raises concerns about data management through the API, potentially impacting privacy and security. In contrast, Llama's ability to run on local machines provides a cost-effective and more secure alternative.

## 6 CONCLUSION

In this paper, we explore the problem of Intent Discovery by harnessing the In-Context learning capabilities of Large Language Models like GPT-3.5, GPT-4, and Llama-2. We propose IntentGPT as a general framework for discovering novel user intents, that automatically generates context-aware prompts using LLMs coupled with advanced prompting techniques like Semantic Few-Shot Sampling or Known Intent Feedback. We highlight IntentGPT's outperformance over other unsupervised and semi-supervised methods, which is particularly remarkable considering that previous approaches require larger training data and multi-stage training processes. In contrast, we do not need training and only a few demonstrations. We present exhaustive experimentation in the novel few-shot In-Context Learning setting, and ablations to understand the influence of hyperparameters.

**Ethics Statement** We acknowledge that biases in user utterances may persist or amplify our model's inferred intents, as we do not explicitly eliminate these biases. We depend on red-teaming efforts to identify and address bias in LLMs. While biases and harmful content filtering are well-addressed on GPT models, concerns persist with LLMs like Llama due to differences in data, training, and deployment techniques, potentially leading to unfair content generation. Mitigating these concerns is a crucial aspect of responsible AI research and development.

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

# A  APPENDIX

## A.1  INTENTGPT ALGORITHM

We define the algorithm of the main operations of IntentGPT in Algorithm A.1. Our proposed system performs Few-Shot Intent Discovery by selecting relevant few-shot examples from the training dataset using Semantic Few-Shot Sampling (SFS). It then determines a subset of intents to include in the prompt, considering both known intents and potential new ones, by employing the Known Intent Feedback (KIF) mechanism. Subsequently, it generates a contextually informative prompt for the Intent Discovery task. The pre-trained Large Language Model (LLM), such as GPT-4, processes this prompt and provides a response parsed to extract predicted intents. If a predicted intent is not already known, it is added to the list of known intents, contributing to the system's ability to discover new intents over time. This algorithm demonstrates the core operations of IntentGPT for Few-Shot Intent Discovery.

---

**Algorithm 1** Few Shot Intent Discovery Algorithm

---

 1: **procedure** FEW SHOT INTENT DISCOVERY($text, trainSet, knownIntents, LLM$)
 2:     $fewShotPool \leftarrow$ SFS($text, trainSet$)                    ▷ Select few-shot examples
 3:     $intentList \leftarrow$ SKIF($text, knownIntents$)               ▷ Select subset of intents to include
 4:     $prompt \leftarrow$ CreatePrompt($fewShotPool, intentList$)
 5:     $response \leftarrow$ LLM($prompt$)
 6:     $predictedIntent \leftarrow$ DiscoverNewIntent($response$)   ▷ Parse response for predicted intents
 7:     **if** $predictedIntent \notin knownIntents$ **then**      ▷ Check if the predicted intent is new to add.
 8:         $knownIntents.add(predictedIntent)$
 9:     **end if**
10: **end procedure**

---

## A.2  PROMPT DESIGN

In this section, we show the prompts we used to empower the in-context learning capabilities of IntentGPT. As described in section 3, we use GPT-4 for designing context-aware prompts. Prompt A is the initial prompt that describes the task of designing prompts using several context examples. We inject $x = 2$ random samples per known intent to give context to GPT-4. The generated prompt, e.g., Prompt B or C, is used as a guideline for Intent Discovery during inference on the LLM tasked at Intent Discovery. Prompt D shows a complete interaction with GPT-4 for an iteration step of Intent Discovery. Note that the prompt contains a list of the known intents, few-shot demonstrations, and a batch of test examples. Prompt E shows a simple human-generated prompt.

Prompt 1: A. Prompt used for In-Context Prompt Generation (ICPG) on GPT-4

```
You are a helpful assistant and an expert in natural language processing.
   You specialize in intent detection and discovery, the task of
   assigning textual utterances to specific intents, some of which are
   pre-defined and others are not and have to be created. Keep in mind
   that in the problem of intent discovery, you need to be aware of the
   known intents and reuse them as much as possible, but need to create
   new intents when there are not known intents that fit the given
   utterance, and never assign a utterance to 'unknown'. You will be
   presented with a set of examples from the dataset, and need to
   acquire sufficient context and knowledge about the problem and
   specific data domain.

As an expert, provide a detailed prompt for an AI language model to solve
   the task of intent discovery, maximizing the model's performance in
   the task. Provide effective guidelines about how to solve the task in
   the prompt, but keep it very concise.

EXAMPLES: {train_examples}
You must respond using the following format:
PROMPT: <prompt>
```

Prompt 2: B. Generated Prompt by GPT-4 as ICPG on CLINC (KIR=0.75)

```
AI language model, your task is to assign the correct intent to a given
    textual utterance. The intent can be one of the pre-defined intents
    or a new one that you create based on the context and knowledge about
     the problem and specific data domain. You should never assign an
    utterance to 'unknown'.

For each utterance, analyze the context and the specific request or
    action implied. If the utterance matches a known intent, assign it to
     that intent. If it doesn't match any known intent, create a new
    intent that accurately represents the request or action implied by
    the utterance.

Remember, the goal is to understand the user's intent as accurately as
    possible. Be aware of the known intents and reuse them as much as
    possible, but don't hesitate to create new intents when necessary.

Here are some examples:

- If the utterance is "how are you named", the intent is "
    what_is_your_name".
- If the utterance is "i need to know the total calories for a chicken
    caesar salad", the intent is "calories".
- If the utterance is "please pause my banking actions", the intent is "
    freeze_account".

Use these examples as a guide, but remember that the utterances can vary
    greatly in structure and content. Your task is to understand the
    underlying intent, regardless of how the utterance is phrased.
```

Prompt 3: C. Generated Prompt by GPT-4 as ICPG on BANKING (KIR=0.75)

```
As an AI language model, your task is to assign the correct intent to a
    given textual utterance. The intents are related to various banking
    and financial services scenarios. You should be aware of the known
    intents and reuse them as much as possible. However, if there is no
    known intent that fits the given utterance, you should create a new
    intent. Remember, you should never assign an utterance to 'unknown'.

The utterances can be questions, statements, or requests related to
    banking services like card transactions, account top-ups, refunds,
    identity verification, card delivery, transfer fees, and more. Your
    task is to understand the context and the specific intent behind each
     utterance.

For example, if the utterance is "what happened to the money after i put
    in the wrong info and it got declined", the intent should be "
    topping_up_by_card". If the utterance is "i cannot see a refund in my
     account", the intent should be "Refund_not_showing_up".

Please note that the same utterance can have different intents based on
    the context. For instance, "my card is expiring, how do i get a new
    one?" has the intent "card_about_to_expire", while "i have enough
    money in my account, so why is my card being declined?" has the
    intent "declined_card_payment".

Your responses should be concise, accurate, and contextually relevant.
```

Prompt 4: D. Complete interaction with IntentGPT Intent Predictor on CLINC

```
You are a helpful assistant and an expert in natural language processing
    and specialize in the task of intent detection.
```

```
AI language model, your task is to assign the correct intent to a given
    textual utterance. The intent can be one of the pre-defined intents
    or a new one that you create based on the context and knowledge about
     the problem and specific data domain. You should never assign an
    utterance to 'unknown'.

For each utterance, analyze the context and the specific request or
    action implied. If the utterance matches a known intent, assign it to
     that intent. If it doesn't match any known intent, create a new
    intent that accurately represents the request or action implied by
    the utterance.

Remember, the goal is to understand the user's intent as accurately as
    possible. Be aware of the known intents and reuse them as much as
    possible, but don't hesitate to create new intents when necessary.

Here are some examples:

- If the utterance is "how are you named", the intent is "
    what_is_your_name".
- If the utterance is "i need to know the total calories for a chicken
    caesar salad", the intent is "calories".
- If the utterance is "please pause my banking actions", the intent is "
    freeze_account".

Use these examples as a guide, but remember that the utterances can vary
    greatly in structure and content. Your task is to understand the
    underlying intent, regardless of how the utterance is phrased.
Make sure each intent is only between one and three words, and as short
    and reusable as possible. Use the same format as the context examples
    . Don't classify the examples below CONTEXT EXAMPLES. Only classify
    the test examples below TEST EXAMPLES. You are prohibited to assign
    intents to 'unknown'. Instead, create a new intent.

Don't discover a new intent if you have already discovered one that is
    similar. Make sure that the intents are not very generic, you can be
    fine-grained. Use the following list of known intents to keep
    reference, reuse them as much as possible:

no, what_is_your_name, calories, shopping_list, freeze_account,
    pto_request_status, current_location, where_are_you_from, income, gas
    , confirm_reservation, maybe, improve_credit_score, book_hotel,
    repeat, apr, damaged_card, tire_pressure, balance, share_location,
    what_are_your_hobbies, insurance_change, car_rental, smart_home,
    gas_type, yes, pto_used, replacement_card_duration, order_status,
    cancel, restaurant_suggestion, rollover_401k, change_accent,
    redeem_rewards, credit_score, reminder, restaurant_reviews,
    meeting_schedule, meal_suggestion, exchange_rate, directions,
    flight_status, calendar, do_you_have_pets, alarm, travel_suggestion,
    update_playlist, ingredients_list, travel_notification,
    what_can_i_ask_you, w2, report_lost_card, book_flight, distance,
    thank_you, travel_alert, calculator, make_call, roll_dice,
    pto_balance, how_old_are_you, international_visa, how_busy, time,
    are_you_a_bot, timezone, change_user_name, mpg, insurance, payday,
    vaccines, fun_fact, report_fraud, pto_request, taxes,
    restaurant_reservation, measurement_conversion, last_maintenance,
    play_music, application_status, credit_limit_change, change_speed,
    date, who_made_you, pin_change, spending_history, definition,
    reminder_update, change_ai_name, tire_change, order, account_blocked,
     calendar_update, routing, cook_time, food_last, interest_rate,
    greeting, user_name, todo_list, ingredient_substitution,
    schedule_maintenance, shopping_list_update, transactions,
    rewards_balance, credit_limit, carry_on, expiration_date,
    change_language, text, next_holiday, who_do_you_work_for
```

```
CONTEXT EXAMPLES:

Utterance: please remind me later, Intent: reminder_update
Utterance: i want to know what health plan i'm currently on, Intent:
    insurance
Utterance: do you work for another individual, Intent:
    who_do_you_work_for
Utterance: tell me my health plan, Intent: insurance
Utterance: is there an insurance plan for my health, Intent: insurance
Utterance: are you working for another person or entity, Intent:
    who_do_you_work_for
Utterance: i'm ready to put in the order for everything on my shopping
    list, Intent: order
Utterance: get everything on my shopping list, Intent: order
Utterance: i'd like to you order everything that's on my shopping list,
    Intent: order
Utterance: it is damaging to my credit score if i miss one credit card
    payment, Intent: improve_credit_score

RESPONSE FORMAT:

ID: , Utterance: <content>, Intent: <intent> Use the same ID in the
    test example.

TEST EXAMPLES (to predict):

ID:0, is there another person you work for
ID:1, does tartine in san francisco take reservations
ID:2, what is the traffic like on the road to downtown dallas
ID:3, what do you think the meaning of life is
ID:4, what is the square root of 80
ID:5, is the friday's full after 4
ID:6, i'm going to need an uber for 4 people to go to patty's bar
ID:7, place an order for my entire shopping list
ID:8, what is the traffic like on the way to town
ID:9, please spell doctor
ID:10, tell my bank i'm in canada the entire week of the 6th to the 13th
ID:11, i need help with some fun tourist things to do in tokyo
ID:12, please remind me at a later time
ID:13, what specific health plan do i have
ID:14, can i harm my credit if i miss a car payment
ID:15, please call mark

Your response:

ID:0, Utterance: is there another person you work for, Intent:
    who_do_you_work_for
ID:1, Utterance: does tartine in san francisco take reservations, Intent:
     restaurant_reservation
ID:2, Utterance: what is the traffic like on the road to downtown dallas,
     Intent: traffic_status
ID:3, Utterance: what do you think the meaning of life is, Intent:
    philosophical_question
ID:4, Utterance: what is the square root of 80, Intent: calculator
ID:5, Utterance: is the friday's full after 4, Intent:
    restaurant_busy_status
ID:6, Utterance: i'm going to need an uber for 4 people to go to patty's
    bar, Intent: book_ride
ID:7, Utterance: place an order for my entire shopping list, Intent:
    order
ID:8, Utterance: what is the traffic like on the way to town, Intent:
    traffic_status
ID:9, Utterance: please spell doctor, Intent: spelling
ID:10, Utterance: tell my bank i'm in canada the entire week of the 6th
    to the 13th, Intent: travel_notification
```

```
ID:11, Utterance: i need help with some fun tourist things to do in tokyo
    , Intent: travel_suggestion
ID:12, Utterance: please remind me at a later time, Intent:
    reminder_update
ID:13, Utterance: what specific health plan do i have, Intent: insurance
ID:14, Utterance: can i harm my credit if i miss a car payment, Intent:
    improve_credit_score
ID:15, Utterance: please call mark, Intent: make_call
```

Prompt 5: E. Basic human-generated task description

```
You are a helpful assistant and an expert in natural language processing
    and specialize in the task of intent detection. Your task is to
    assign utterances to their corresponding intent. The intent can be
    one of the pre-defined intents or a new one that you create based on
    the context and knowledge about the problem and specific data domain.
     You should never assign an utterance to 'unknown'. For each
    utterance, analyze the context and the specific request or action
    implied. If the utterance matches a known intent, assign it to that
    intent. If it doesn't match any known intent, create a new intent
    that accurately represents the request or action implied by the
    utterance. Remember, the goal is to understand the user's intent as
    accurately as possible. Be aware of the known intents and reuse them
    as much as possible, but don't hesitate to create new intents when
    necessary.
```

## A.3   DETAIL ON OFF-THE-SHELF LLMS USED IN INTENTGPT

This section describes our usage of pre-trained language models. Find here details on specific models, ways to access them, and estimated costs for running them. For GPT-4 and GPT-3.5 we use the available endpoints at OpenAI[3], namely `gpt-3.5-turbo` and `gpt-4`. For Llama-2, we use the AnyScale[4] endpoint that uses `meta-llama/Llama-2-70b-chat-hf`. Table A.3 displays relevant information about the models, the tokens required for a single query with a batch of 16 samples, and their monetary costs. Finally, we use a pre-trained SentenceBert (Reimers & Gurevych, 2019) transformer (`all-MiniLM-L6-v2`) model aimed at extracting text embeddings for computing text similarities and clustering.

Considering a batch size of 16 samples for query, CLINC requires 141 iterations to perform a complete test evaluation, while BANKING requires 193. For instance, Table A.3 suggests that an evaluation run using GPT-4 costs approximately 7$ on CLINC and 9.65$ on BANKING. In terms of timing, we measure that GPT-3.5 takes approximately 20 minutes, while GPT-4 finishes in around 2 hours. This results show that running inference on this models can be costly, but presents a reasonable alternative to fine-tuning a model on specific cases and domains.

| Model | Endpoint API | # Tokens in Prompt | Cost/Batch ($) |
|---|---|---|---|
| gpt3.5-turbo | OpenAI | 1240 | 0.002 |
| gpt-4 | OpenAI | 1240 | 0.05 |
| Llama-2-70b-chat-hf | AnyScale | 1518 | 0.0015 |

Table 4: Cost per batch inference for different models.

## A.4   INFLUENCE OF $K$ FOR K-MEANS CLUSTERING

We explore the choice of $K$ in K-Means under the KIR=0.75 setting. We explore all IntentGPT models in the 50 shot setting. Results are shown in Figures 5 and 6. We observe that there exists an optimal range where the models perform at their best, typically aligning with the ground truth

---

[3]https://platform.openai.com/playground
[4]https://app.endpoints.anyscale.com/

number of intents in the datasets, approximately $K = 150$ for CLINC and $K = 77$ for BANKING. It's worth noting that GPT models tend to saturate in metric performance as $K$ increases, while Llama-2 exhibits a slight decline.

In our IntentGPT experiments, we automatically determine the value of $K$ for K-Means by first using DBSCAN on predicted intent embeddings. This approach aligns with real-world scenarios where the total number of intents is unknown. We find that $K$ tends to be approximately 155 for CLINC and 88 for BANKING.

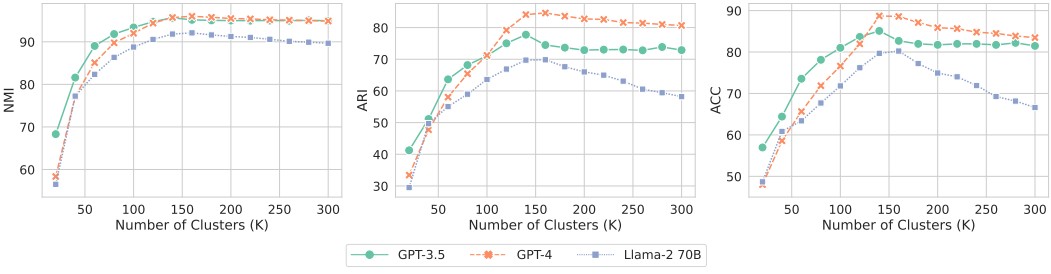

Figure 5: Analyzing the influence of $K$ for K-Means on CLINC dataset (150 intents).

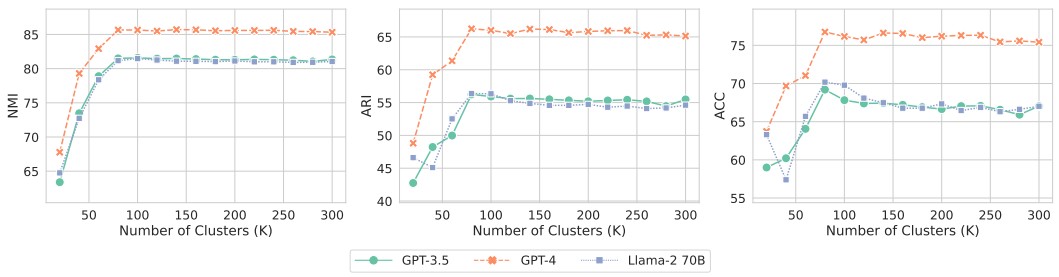

Figure 6: Analyzing the influence of $K$ for K-Means on BANKING (77 intents).

### A.5 SAMPLING TEMPERATURE OF LLMS

The sampling temperature quantifies the desired level of stochasticity during the generation process, typically represented as a positive scalar within the 0 to 2 range. Higher temperatures foster greater creativity in the model, while lower temperatures favor determinism. In our specific context, our objective is to foster creativity, enabling the model to uncover novel and precise intents while maintaining a connection with existing intents.

In our evaluation, depicted in Figures 7 and 8, we observe a trend of declining metrics as temperature increases. This phenomenon occurs because the model's increased randomness at higher temperatures can lead to unintended variations in the generated intents, potentially affecting the accuracy of the model's responses. Thus, selecting an optimal temperature is crucial to strike the right balance between creativity and coherence in intent generation.

### A.6 ARE THE LLMS WE USE PRE-TRAINED ON THE DATASETS?

One could argue that the effectiveness of IntentGPT arises from the underlying LLM, such as GPT-4, having been trained on the data (Mireshghallah et al., 2022; Mattern et al., 2023). Nevertheless, addressing this question proves challenging due to a lack of transparency in this matter. To explore this further, we will conduct three analyses. First, we will perform a series of interventions on GPT-4 to determine its familiarity with the CLINC and BANKING test datasets. Second, we will revisit our ablation study on GPT-4 to gain insights into the source of improvements. Finally, we introduce a new metric to assess the similarity between the intents generated by IntentGPT using GPT-4 and the ground truth.

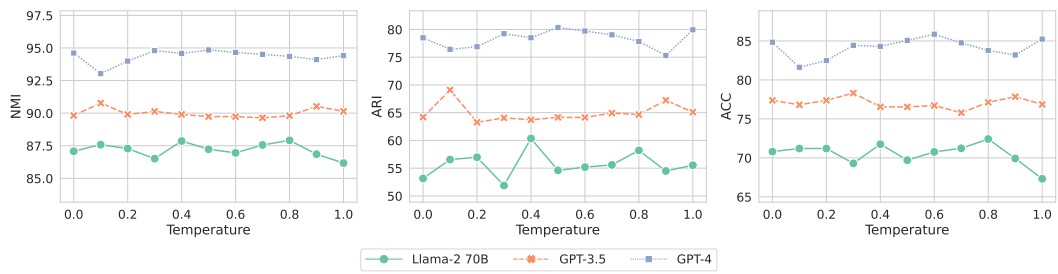

Figure 7: Analyzing the influence of the LLM temperature on CLINC dataset.

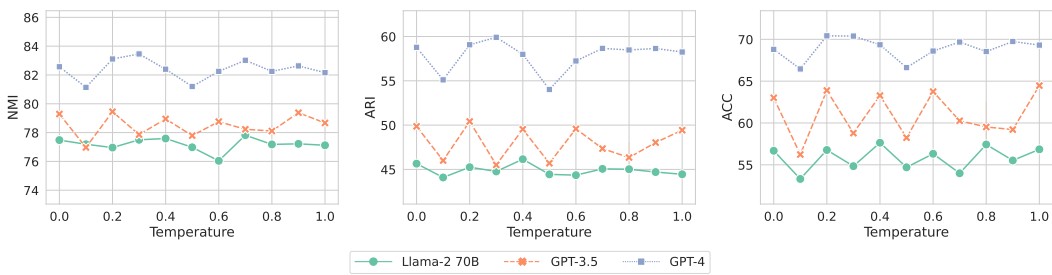

Figure 8: Analyzing the influence of the LLM temperature on BANKING dataset.

**GPT-4 on Intent Discovery Datasets.** We perform a series of interventions and engage in conversation with the GPT-4 model, seeking to understand how much it knows from CLINC and BANKING datasets. We ask for detailed descriptions of the datasets, generation of train and test examples, and prediction of test labels. From this study, we acknowledge that GPT-4 knows both datasets, domain contexts, number of samples and splits, and understands the task. However, we did not find that any of the examples generated were part of the datasets, hence the examples were hallucinated.

**Further discussion on ablation study.** We revisit the ablation study from Table 2 to understand where the gains in performance come from. The most simple set-up, where we only include a human-generated task description (first row in the table), shows very poor performance in all metrics. This shows that the model by itself is not able to perform well in the task. Is when we add the proposed techniques such as Known Intent Feedback (KIF), Semantic Few-Shot (SFS), or In-Context Prompt Generation (ICP) that the models start to achieve state-of-the-art performance.

**Measuring the similarity between generated intents and real ones.** We want to measure the similarity between the predicted and the ground truth intents, to learn if GPT-4 has memorized the test labels during training with the test set. In contrast, we would like the model to generalize enough that it creates novel intents and consistently assigns them to the corresponding utterance. To that end, we design the *Frechet Bert Distance* (FBD) metric that takes inspiration from the well-known Fréchet Inception Distance Heusel et al. (2017) used in computer vision for assessing the quality of image generative models. We build on top of Semeniuta et al. (2018) and choose to use SBERT embeddings for computing the similarity between text distributions in the latent space.

Having two sets of text samples, $s_1$ and $s_2$, we extract SBERT embeddings and compute the Fréchet Distance between them. If both sets are equal, we will measure zero, and if they are completely different we will measure close to one. For example, having $s_1 = ('a', 'b', 'c')$ and $s_2 = ('d', 'e', 'f')$, gives $\text{FBD}(s_1, s_2) = 0.96$.

For our experiment, we select the set of discovered intents $I_d$ (excluding the known intents at the start) and the set of ground truth unknown intents to discover $I_u$, and compute SBERT embeddings. Moreover, we compute the Fréchet Distance between both sets of embeddings. Our final measurement for the results using GPT-4 on CLINC with 50-shots is $\text{FDB}(I_d, I_u) = 0.54$. In the case of BANKING we obtain $\text{FDB}(I_d, I_u) = 0.55$, which suggests that generated intents are sufficiently

different in the used datasets. Additionally, we can evaluate the generated intents qualitatively using Table A.7, which shows also the differences between the generated and real intents.

This is a relevant problem and orthogonal to many research directions in LLMs because pre-trained models that are trained on test sets, may break the confidence of future research efforts in popular benchmarks for important problems. Future work must put effort into detecting which models can be evaluated in which dataset, and even though that is the case, evaluate the level of memorization of test predictions and model generalization.

### A.7 QUALITATIVE ANALYSIS OF THE DISCOVERED INTENTS

The design of IntentGPT allows us to qualitatively assess the discovered intents because the model will inherently find the need for a new intent and create a name for it. Table A.7 displays some of the examples seen during test on CLINC with GPT-4 on KIR=0.75 and 50 shot settings. By looking at the test predictions we observe that the model is consistent in assigning the utterances to the same intent, even though the name of the intent does not correspond the the ground truth. This is the correct behavior, because clustering metrics do not take into account that discovered and ground truth intents are the same. However, we can see errors in utterances with subtle nuances like the difference between "bill inquiry" and "bill payment", which was the prediction of IntentGPT for "pay bill". Also, "device_pairing" is different than the ground truth "sync_device", but is consistently assigned to the corresponding utterances.

Furthermore, we compute SBert embeddings of both discovered intents and ground truth intents and compute a TSNE visualization into a 2D grid. Results are shown in Figure 9. This visualization displays that many intents coincide while many others are semantically different. We consider that this is consistent with the Frechet Bert Distance score obtained before and that both sets are sufficiently different.

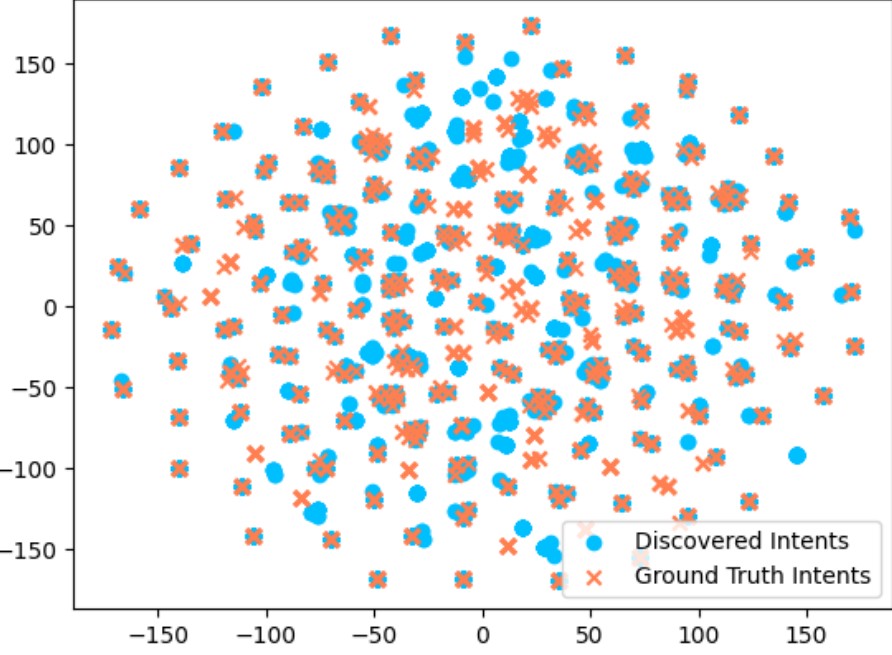

Figure 9: TSNE visualization of S-Bert embeddings of discovered intents and ground truth intents.

| Utterance | Predicted Intent | Ground Intent |
|---|---|---|
| i lost my phone and need help retrieving it | device_location | find phone |
| are you able to track a phone | device location | find phone |
| call my phone, i can't find it | make call | find phone |
| at what time will my flight land | flight status | flight status |
| is flight dl123 coming in on time | flight status | flight status |
| i would like to know when my flight scheduled to board | flight status | flight status |
| choose heads or tails and tell me what you came up with | flip coin | flip coin |
| lets do a coin toss | roll dice | flip coin |
| what's the least payment i can make on my cable bill | bill_inquiry | min_payment |
| how much is the least i can pay for power bill | bill_inquiry | min_payment |
| i need to pay the smallest amount on my phone bill | bill_inquiry | min_payment |
| i would like to know more about getting a new credit card | credit_card_inquiry | new_card |
| what's the procedure to apply for a new credit card | credit_card_inquiry | new_card |
| i would like to get a new credit card | credit_card_inquiry | new_card |
| does bank of america give credit cards to people like me | credit_card_inquiry | new_card |
| can you pair with my phone | device_pairing | sync_device |
| my phone needs to be unsynced now | device_pairing | sync_device |
| disconnect from my phone please | device_pairing | sync_device |
| you need to connect to my phone | device_pairing | sync_device |
| when should i change my oil | schedule maintenance | oil change when |
| when should i get my next oil change | schedule maintenance | oil change when |
| when is the best time for my next oil change | schedule maintenance | oil change when |
| pay the cable bill with my visa card | bill inquiry | pay bill |
| my electric bill should be paid today | bill inquiry | pay bill |
| i need to pay my water bill | bill payment | pay bill |
| pay my cable bill from my checking account | bill payment | pay bill |
| pay my insurance bill | bill payment | pay bill |
| i think someone stole my card and used it | report fraud | report fraud |
| i have to report fraudulent activity on my bank of the west card | report fraud | report fraud |
| i need to tell you about my lost card | report lost card | report lost card |
| i need to report my stolen card | report lost card | report lost card |
| my capital one credit card was stolen | report lost card | report lost card |
| i require a car maintenance | schedule maintenance | schedule maintenance |
| i got to schedule some car maintenance | schedule maintenance | schedule maintenance |
| add a new meeting with tom for 6pm | meeting schedule | schedule meeting |
| would you schedule a meeting with carrie and lisa please | meeting schedule | schedule meeting |

Table 5: Intents generated by IntentGPT using GPT-4 and 50 shots in CLINC test set.

| Model | NMI | ARI | ACC |
|---|---|---|---|
| **MTOP English** | | | |
| IntentGPT-3.5 | 83.54 | 80.43 | 76.47 |
| IntentGPT-4 | 87.98 | 82.22 | 85.09 |
| **MTOP Spanish** | | | |
| IntentGPT-3.5 | 88.21 | 88.74 | 82.42 |
| IntentGPT-4 | 90.37 | 90.90 | 85.69 |
| **MTOP French** | | | |
| IntentGPT-3.5 | 88.97 | 88.44 | 83.43 |
| IntentGPT-4 | 84.34 | 80.62 | 76.69 |
| **MTOP German** | | | |
| IntentGPT-3.5 | 80.44 | 65.82 | 67.26 |
| IntentGPT-4 | 86.64 | 81.87 | 77.54 |
| **MTOP Hindi** | | | |
| IntentGPT-3.5 | 60.43 | 36.20 | 43.33 |
| IntentGPT-4 | 73.23 | 66.36 | 66.01 |
| **MTOP Thai** | | | |
| IntentGPT-3.5 | 28.38 | 5.66 | 20.66 |
| IntentGPT-4 | 23.72 | 7.42 | 17.67 |

Table 6: Results on MTOP dataset with 6 different languages.

## A.8    MULTILINGUAL DATASETS

We evaluate IntentGPT's robustness when being used for languages other than English. To that end, we use MTOPLi et al. (2020) dataset, composed of utterances and intents in six different languages: English, German, French, Spanish, Hindi, and Thai. We evaluate IntentGPT in a 10-shot setting and using KIR=0.75. Results are shown in table A.8

We observe that IntentGPT is able to generalize to languages other than English without requiring extra human effort. The first LLM in charge of generating the prompt (ICPG) seems to perform well at automatically generating the prompt in the desired language (see the generated prompts in Spanish and French).

Quantitative results show that our method works well in this multilingual setting, showing metrics in a similar range to the ones seen in other benchmarks, except for the case of Thai, where the GPT3.5 and GPT4 models struggle to handle. We leave as future work the comparison of IntentGPT with existing methods in this benchmark since existing methods do not report results on these datasets. We believe that these results reinforce the idea that IntentGPT is a robust approach for open-world class discovery tasks, and the novelty of the ICP and few-shot sampling techniques proposed.

Prompt 6: Prompt generated by GPT-4 ICPG on MTOP Spanish

```
PROMPT: Para realizar la tarea de deteccion y descubrimiento de
    intenciones, debes analizar cada enunciado textual y asignarlo a una
    intencion especifica. Algunas de estas intenciones ya est n
    predefinidas, mientras que otras no y tendr s que crearlas. Debes
    tener en cuenta las intenciones conocidas y reutilizarlas tanto como
    sea posible, pero tambi n debes estar dispuesto a crear nuevas
    intenciones cuando no existan intenciones conocidas que se ajusten al
     enunciado dado. Nunca asignes un enunciado a 'desconocido'. Por
    ejemplo, si el enunciado es "comparte share psychedelics y social
    justice con emery", la intenci n ser a IN:SHARE_EVENT. Si el
    enunciado es "  puedes contestar la llamada de austin pate?", la
    intenci n ser a IN:ANSWER_CALL. Es importante que adquieras
```

| Method | Embedding type | CLINC | | | BANKING | | |
|---|---|---|---|---|---|---|---|
| | | NMI ↑ | ARI↑ | ACC↑ | NMI ↑ | ARI↑ | ACC↑ |
| IntentGPT-3.5 | SentenceBert | 91.66 | 71.58 | 78.99 | 88.21 | 88.74 | 82.42 |
| IntentGPT-3.5 | text-embedding-ada-002 | 88.64 | 61.42 | 71.36 | 69.46 | 39.51 | 52.12 |
| IntentGPT-4 | SentenceBert | 94.01 | 77.17 | 83.96 | 91.66 | 71.58 | 78.99 |
| IntentGPT-4 | text-embedding-ada-002 | 90.73 | 65.99 | 74.89 | 88.64 | 61.42 | 71.36 |

Table 7: Results of using different types of embeddings for clustering intents

```
suficiente contexto y conocimiento sobre el problema y el dominio de
datos espec fico. Como experto en procesamiento de lenguaje natural,
 debes ser capaz de interpretar el significado subyacente de cada
enunciado y asignarlo a la intenci n correcta. Recuerda que el
conjunto de datos puede estar en diferentes idiomas adem s del
ingl s. Por lo tanto, debes responder con un aviso en el mismo
idioma que los ejemplos de contexto.
```

Prompt 7: Prompt generated by GPT-4 ICPG on MTOP French

```
PROMPT: Pour chaque phrase ou texte donn , votre t che est de
  d terminer l\'intention qui se cache derri re. L\'intention est
  g n ralement li e   une action que l\'utilisateur souhaite
  accomplir ou   une information qu\'il cherche   obtenir. Par
  exemple, si l\'utilisateur dit "repetez cette chanson encore une fois
  ", l\'intention serait "IN:REPLAY_MUSIC", indiquant que l\'
  utilisateur souhaite rejouer la chanson actuelle. \n\nIl est
  important de noter que certaines intentions peuvent  tre
  r utilis es pour diff rentes phrases. Par exemple, "repetez cette
  chanson encore une fois" et "repassez la chanson actuelle" ont tous
  deux l\'intention "IN:REPLAY_MUSIC". Cependant, si une phrase ne
  correspond   aucune intention connue, vous devrez cr er une
  nouvelle intention. \n\nIl est  galement  crucial de ne jamais
  attribuer une phrase   l\'intention \'inconnue\'. Chaque phrase a
  une intention, m me si elle n\'est pas imm diatement  vidente . \n\
  nLors de l\'attribution des intentions, veillez    prendre en compte
  le contexte de la phrase et   utiliser vos connaissances sur le
  domaine sp cifique des donn es. Par exemple, si l\'utilisateur dit
  "ou se deroulent la plupart de mes rappels?", l\'intention serait "IN
  :GET_REMINDER_LOCATION", indiquant que l\'utilisateur cherche
  savoir o  se d roulent la plupart de ses rappels. \n\nEnfin, n\'
  oubliez pas que le format de la r ponse n\'est pas sp cifi . Vous
  pouvez donc r pondre de la mani re qui vous semble la plus
  appropri e, tant que vous identifiez correctement l\'intention.
```

## B  COMPUTATION OF INTENT EMBEDDINGS FOR CLUSTERING

We compute embeddings of predicted intents to perform clustering. Here we show an ablation study to decide the best type of embeddings for our task. We experiment with OpenAI embeddings `text-embedding-ada-002` [5]. Results are shown in Table 7. We find that SentenceBert outperforms OpenAI embeddings. The most likely reason for OpenAI embeddings being worse is that it is primarily used to encode large contexts like documents for document retrieval, whereas SentenceBert is trained to efficiently encode short contexts, or sentences, like the size of our intents.

## C  RESULTS ON ADDITIONAL DATASETS: SNIPS AND STACKOVERFLOW

We report results on two additional datasets, namely SNIPS Coucke et al. (2018) and StackOverflow Xu et al. (2015). These are datasets that contain a much-reduced number of intents compared

---

[5]https://platform.openai.com/docs/guides/embeddings

| Method | KIR | SNIPS | | | StackOverflow | | |
|---|---|---|---|---|---|---|---|
| | | NMI ↑ | ARI↑ | ACC↑ | NMI ↑ | ARI↑ | ACC↑ |
| DSSCC | 0.75 | 90.44 | 89.03 | 94.87 | 77.08 | 68.67 | **82.65** |
| CDAC+ | 0.75 | 89.30 | 86.82 | 93.63 | 69.46 | 52.59 | 73.48 |
| IntentGPT-3.5 (ours) | 0.75 | 85.26 | 80.48 | 89.14 | 80.56 | 71.57 | 81.69 |
| IntentGPT-4 (ours) | 0.75 | **91.42** | **90.63** | **94.89** | **81.78** | **76.85** | 81.75 |

Table 8: Results on SNIPS and StackOverflow datasets.

to CLINC or BANKING, therefore we can judge the robustness of IntentGPT in this regime. We make use of 10-shot and KIR=0.75. Results are shown in Table 8.

