# OpenReview forum: "IntentGPT: Few-Shot Intent Discovery with Large Language Models"
_ICLR.cc/2024/Conference — Submitted to ICLR 2024_

### Official Review · Reviewer_vjHF · 2023-10-29

**Soundness:** 2 fair
**Presentation:** 2 fair
**Contribution:** 2 fair
**Rating:** 5
**Confidence:** 5

**Summary:**

This paper presents a framework for Intent Discovery by utilizing pre-trained LLM s like GPT 4 for few shot in context learning. The authors performed prompt engineering to guide LLMs to produce the desired generations. Experimental results show improvement over existing approaches.

**Strengths:**

Good prompt engineering to guide GPT-4 to produce intents.

Incorporation of few shot examples to improve performance.

Improving performance without training overhead.

**Weaknesses:**

Tested on only 2 datasets; difficult to not judge robustness of the proposed method without seeing results on different datasets.

Prompt and few shot example usage might change depending on the data.

Known intent feedback might not be usable for zero shot context.

**Questions:**

Please perform experiments on more datasets that have diverse input structure. (Many datasets are out there: https://paperswithcode.com/task/intent-detection)

---

> ### Author Response · Authors · 2023-11-23
>
> We appreciate the reviewer's time to read our paper and provide feedback. We are glad that the reviewer found the paper to provide good guidelines for how to prompt GPT-4 and discover intents without any training overhead. We respond to the comments in detail below.
>
> > Tested on only 2 datasets. Please perform experiments on more datasets that have diverse input structure.
>
> We picked those two datasets for our paper because they're widely recognized as the standard benchmarks in open-set intent discovery [3, 4, 5], which makes it fair to compare our work to existing methods. However, the comment inspired us to extend our results to three new datasets. First, as was highlighted by reviewer kRkh, we have run our method on a multilingual dataset (MTOP) [6] and presented the results in our response to them. Please see our comment there for results on the MTOP dataset. To further strengthen our results we consider two additional datasets, namely SNIPS [1], and StackOverflow [2]. These datasets can be interesting because they show a much reduced number of categories, hence will allow us to judge the robustness of our model in a regime with a small number of categories. The results are shown as follows:
>
>
> | Model                | KIR | NMI   | ACC   | ARI   |
> |----------------------|-----|-------|-------|-------|
> | **SNIPS dataset**     |     |       |       |       |
> | DSSCC     | 0.75 | 90.44 | 94.87 | 89.03 |
> | CDAC+     | 0.75 | 89.30 | 93.63 | 86.82 |
> | IntentGPT3.5 (10 shot) (ours)    | 0.75| 85.26 | 89.14 | 80.48  |
> | IntentGPT-4 (10 shot) (ours)       | 0.75| **91.42** | **94.89** | **90.63**  |
> | **StackOverflow dataset**     |     |       |       |       |
> | DSSCC     | 0.75 | 77.08|**82.65**|68.67  |
> | CDAC+     | 0.75 | 69.84|73.48|52.59  |
> | IntentGPT3.5 (10 shot) (ours)      | 0.75| 80.56|81.69|71.57  |
> | IntentGPT-4 (10 shot)   (ours)     | 0.75| **81.78**|81.75|**76.85**  |
>
> We compare IntentGPT-3.5 and IntentGPT-4 against available public results on two of the baselines presented in the paper, namely DSSCC [7] and CDAC+ [8]. Our model outperforms these methods except DSSCC on StackOverflow, which performs better for the ACC metric. These results required no human intervention in modifying the prompts, or training, showcasing the robustness of IntentGPT.
>
> > Prompt and few shot example usage might change depending on the data.
>
> While the IntentGPT prompt and few-shot examples might change across datasets our model is able to automatically adapt the prompts and in-context examples to a given dataset, hence generating more informative and valuable prompts. Furthermore, it does not require human effort when porting the model into a new dataset, as it is fully automated for generating the needed prompts given the new dataset, capturing the relevant nuances about the data domain.
>
> > Known intent feedback might not be usable for zero shot context.
>
> Our Known Intent Feedback technique is indeed usable in a zero-shot setting. In this setting, we inject the discovered intents back into the prompt, starting from an empty database. In fact, we have experiments available on the paper for the zero-shot setting case. We show the results of our method in Table 1 on the paper (see results for the unsupervised setting), and also in the first row of ablations in Table 2, for both GPT3.5 and GPT-4 on the two datasets.

---

> > ### Author Response · Authors · 2023-11-23
> > **References**
> >
> > [1] Coucke, Alice, et al. "Snips voice platform: an embedded spoken language understanding system for private-by-design voice interfaces." arXiv preprint arXiv:1805.10190 (2018).
> >
> > [2] Jiaming Xu, Peng Wang, Guanhua Tian, Bo Xu, Jun Zhao, Fangyuan Wang, and Hongwei Hao. 2015. Short text clustering via convolutional neural networks. In Proceedings of the 1st Workshop on Vector Space Modeling for Natural Language Processing, pages 62–69, Denver, Colorado. Association for Computational Linguistics
> >
> > [3] Hanlei Zhang, Hua Xu, Ting-En Lin, and Rui Lyu. Discovering new intents with deep aligned clustering. In AAAI Conference on Artificial Intelligence, 2021b.
> >
> > [4] Rajat Kumar, Mayur Patidar, Vaibhav Varshney, Lovekesh Vig, and Gautam Shroff. Intent detection and discovery from user logs via deep semi-supervised contrastive clustering. In PConference of the North American Chapter of the Association for Computational Linguistics: Human Language Technologies, 2022.
> >
> > [5] Xiang Shen, Yinge Sun, Yao Zhang, and Mani Najmabadi. Semi-supervised intent discovery with contrastive learning. In Workshop on Natural Language Processing for Conversational AI, 2021.
> >
> > [6] Li, Haoran, et al. "MTOP: A comprehensive multilingual task-oriented semantic parsing benchmark." arXiv preprint arXiv:2008.09335 (2020).
> >
> > [7] Kumar, Rajat, et al. "Intent Detection and Discovery from User Logs via Deep Semi-Supervised Contrastive Clustering." Proceedings of the 2022 Conference of the North American Chapter of the Association for Computational Linguistics: Human Language Technologies. 2022.
> >
> > [8] Lin, Ting-En, Hua Xu, and Hanlei Zhang. "Discovering new intents via constrained deep adaptive clustering with cluster refinement." Proceedings of the AAAI Conference on Artificial Intelligence. Vol. 34. No. 05. 2020.

---

### Official Review · Reviewer_kRkh · 2023-10-31

**Soundness:** 2 fair
**Presentation:** 3 good
**Contribution:** 2 fair
**Rating:** 3
**Confidence:** 4

**Summary:**

The paper proposes IntentGPT, a framework for few-shot intent discovery using large language models (LLMs) like GPT-3.5, GPT-4, and Llama-2.

**Strengths:**

1. Eliminates the need for training and extensive domain-specific data by leveraging the knowledge and generalization capabilities of LLMs through in-context learning.

2. Introduces an automatic prompt generation module that leverages examples from the training set to create high-quality prompts tailored for intent discovery.

3. Proposes techniques like Semantic Few-Shot Sampling to retrieve relevant examples and Known Intent Feedback to reuse discovered intents.

4. Shows competitive performance compared to prior state-of-the-art semi-supervised methods on intent discovery benchmarks like CLINC and BANKING.

5. Provides comprehensive analysis on the influence of various hyperparameters for few-shot in-context learning using frozen LLMs.

**Weaknesses:**

1. The core idea of using LLMs for few-shot learning, while powerful, is not highly creative or novel at this point given extensive prior work on leveraging LLMs. The techniques like prompt engineering and few-shot sampling, while optimized for intent discovery, mostly draw from existing approaches for adapting LLMs.

2. Is there a suspicion of cheating in the KIF mechanism? Is there any unfairness compared to traditional deep learning paradigms?

3. The research is limited to English datasets, however, large language models can be easily extended to other languages, so it is necessary to test the performance of IntentGPT in other languages. I think the author may lack this necessary comparison.

**Questions:**

See Above.

**Details Of Ethics Concerns:**

No.

---

> ### Author Response · Authors · 2023-11-23
>
> Thank you for your thoughtful feedback. We are happy to see that the reviewer sees benefits to our Known Intent Feedback (KIF) mechanism and that we provide a comprehensive analysis of model hyper-parameters. We respond to the reviewer’s comments in detail below.
>
> > The idea of using LLMs for few-shot learning is not highly creative or novel
>
> While LLM-based few-shot learning has received significant attention in recent literature[1, 2], the challenge of generating effective prompts for LLMs remains unsolved [3]. We recognized an opportunity to improve existing methods by proposing two valuable contributions. First, we introduce the use of an In-Context Prompt Generator (ICPG) LLM that can generate dataset-specific and informative prompts (ICP). Second, we introduce the technique of Known Intent Feedback (KIF) that can be extended for the general tasks of open-world class discovery. For the task of Intent Discovery, we see that these techniques can boost performance for our method, outperforming previous baselines using substantially less data, no training overhead, and minimal human effort.
>
> > Is Known Intent Feedback (KIF) unfair to previous deep learning approaches?
>
> We respectfully disagree.  KIF is not unfair compared to previous methods as it does not provide our approach with unfair information. KIF works as a regularization in that it restricts the amount of discovered classes by making sure they are separate enough from each other.
>
> > The research is limited to English datasets. Can IntentGPT work with multilingual datasets?
>
> We chose the English datasets since they seem to be the standard benchmarks in recent papers for the open-set intent discovery task [4, 5, 6].  However, your comment inspired us to test this method in a different language. IntentGPT can be seamlessly used in multiple languages supported by the underlying language model. Therefore,  we ran our method on the MTOP [7] dataset which consists of six languages: English, German, French, Spanish, Hindi, and Thai, and the results are shown below.
>
> Table: Results on MTOP dataset
>
> | Model                | KIR | NMI   | ACC   | ARI   |
> |----------------------|-----|-------|-------|-------|
> | **MTOP English**     |     |       |       |       |
> | GPT3.5 (10 shot)     | 0.75| 83.54 | 76.47 | 80.43 |
> | GPT-4 (10 shot)      | 0.75| 87.98 | 82.22 | 85.09 |
> | **MTOP Spanish**     |     |       |       |       |
> | GPT3.5 (10 shot)     | 0.75| 88.21 | 82.42 | 88.74 |
> | GPT-4 (10 shot)      | 0.75| 90.37 | 85.69 | 90.90 |
> | **MTOP French**      |     |       |       |       |
> | GPT3.5 (10 shot)     | 0.75| 88.97 | 83.43 | 88.44 |
> | GPT-4 (10 shot)      | 0.75| 84.34 | 76.69 | 80.62 |
> | **MTOP German**      |     |       |       |       |
> | GPT3.5 (10 shot)     | 0.75| 80.44 | 67.26 | 65.82 |
> | GPT-4 (10 shot)      | 0.75| 86.64 | 77.54 | 81.87 |
> | **MTOP Hindi**       |     |       |       |       |
> | GPT3.5 (10 shot)     | 0.75| 60.43 | 43.33 | 36.2  |
> | GPT-4 (10 shot)      | 0.75| 73.23 | 66.01 | 66.36 |
> | **MTOP Thai**        |     |       |       |       |
> | GPT3.5 (10 shot)     | 0.75| 28.38 | 20.66 | 5.66  |
> | GPT-4 (10 shot)      | 0.75| 23.72 | 17.67 | 7.42  |
>
> We observe that IntentGPT is able to generalize to languages other than English without requiring extra human effort. The first LLM in charge of generating the prompt (ICPG) seems to perform well at automatically generating the prompt in the desired language (see next comment to observe the prompts generated by ICP in Spanish, French, and Hindi).
>
> Furthermore, quantitative results in the task of Intent Discovery displayed in the table show that our method works well in this multilingual setting, showing metrics in a similar range to the ones seen in other benchmarks, except for the case of Thai, where the GPT3.5 and GPT4 models struggle to handle. We leave as future work the comparison of IntentGPT with existing methods in this benchmark since existing methods do not report results on these datasets. We believe that these results reinforce the idea that IntentGPT is a robust approach for open-world class discovery tasks, and the novelty of the ICP and few-shot sampling techniques proposed. We have incorporated this new setting in the Appendix, including details on the automatically generated prompts.

---

> > ### Author Response · Authors · 2023-11-23
> > **Prompts generated by ICPG in multiple languages (MTOP dataset)**
> >
> > For instance, here is the prompt generated by ICP for MTOP in Spanish:
> >
> > *PROMPT: Para realizar la tarea de detección y descubrimiento de intenciones, debes analizar cada enunciado textual y asignarlo a una intención específica. Algunas de estas intenciones ya están predefinidas, mientras que otras no y tendrás que crearlas. Debes tener en cuenta las intenciones conocidas y reutilizarlas tanto como sea posible, pero también debes estar dispuesto a crear nuevas intenciones cuando no existan intenciones conocidas que se ajusten al enunciado dado. Nunca asignes un enunciado a \'desconocido\'. \n\nPor ejemplo, si el enunciado es "comparte share psychedelics y social justice con emery", la intención sería IN:SHARE_EVENT. Si el enunciado es "¿ puedes contestar la llamada de austin pate?", la intención sería IN:ANSWER_CALL. \n\nEs importante que adquieras suficiente contexto y conocimiento sobre el problema y el dominio de datos específico. Como experto en procesamiento de lenguaje natural, debes ser capaz de interpretar el significado subyacente de cada enunciado y asignarlo a la intención correcta. \n\nRecuerda que el conjunto de datos puede estar en diferentes idiomas además del inglés. Por lo tanto, debes responder con un aviso en el mismo idioma que los ejemplos de contexto.*
> >
> > For French, we obtain the following prompt:
> >
> > *PROMPT: Pour chaque phrase ou texte donné, votre tâche est de déterminer l\'intention qui se cache derrière. L\'intention est généralement liée à une action que l\'utilisateur souhaite accomplir ou à une information qu\'il cherche à obtenir. Par exemple, si l\'utilisateur dit "repetez cette chanson encore une fois", l\'intention serait "IN:REPLAY_MUSIC", indiquant que l\'utilisateur souhaite rejouer la chanson actuelle. \n\nIl est important de noter que certaines intentions peuvent être réutilisées pour différentes phrases. Par exemple, "repetez cette chanson encore une fois" et "repassez la chanson actuelle" ont tous deux l\'intention "IN:REPLAY_MUSIC". Cependant, si une phrase ne correspond à aucune intention connue, vous devrez créer une nouvelle intention. \n\nIl est également crucial de ne jamais attribuer une phrase à l\'intention \'inconnue\'. Chaque phrase a une intention, même si elle n\'est pas immédiatement évidente. \n\nLors de l\'attribution des intentions, veillez à prendre en compte le contexte de la phrase et à utiliser vos connaissances sur le domaine spécifique des données. Par exemple, si l\'utilisateur dit "ou se deroulent la plupart de mes rappels?", l\'intention serait "IN:GET_REMINDER_LOCATION", indiquant que l\'utilisateur cherche à savoir où se déroulent la plupart de ses rappels. \n\nEnfin, n\'oubliez pas que le format de la réponse n\'est pas spécifié. Vous pouvez donc répondre de la manière qui vous semble la plus appropriée, tant que vous identifiez correctement l\'intention.'*
> >
> > And for Hindi, we obtain:
> >
> > *आपको निम्नलिखित उदाहरणों के आधार पर उच्चारण का उद्देश्य समझना होगा और उसे एक विशेष उद्देश्य से जोड़ना होगा। कृपया ध्यान दें कि कुछ उद्देश्य पहले से परिभाषित हो सकते हैं, जबकि अन्य नए हो सकते हैं और उन्हें बनाने की आवश्यकता हो सकती है। अगर कोई उच्चारण किसी मौजूदा उद्देश्य से मेल नहीं खाता है, तो आपको एक नया उद्देश्य बनाना होगा। आपका लक्ष्य होना चाहिए कि आप जितना संभव हो सके, मौजूदा उद्देश्यों का पुन: उपयोग करें, लेकिन जब ऐसा संभव नहीं हो, तो नए उद्देश्य बनाएं। आपको कभी भी उच्चारण को 'अज्ञात' के रूप में नहीं असाइन करना चाहिए। आपका उत्तर निम्नलिखित प्रारूप में होना चाहिए: उद्देश्य: <उद्देश्य>*

---

> ### Author Response · Authors · 2023-11-23
> **References**
>
> [1] Tom Brown, Benjamin Mann, Nick Ryder, Melanie Subbiah, Jared D Kaplan, Prafulla Dhariwal, Arvind Neelakantan, Pranav Shyam, Girish Sastry, Amanda Askell, et al. Language models are few-shot learners. Advances in neural information processing systems (NeurIPS), 2020.
>
> [2] Sang Michael Xie, Aditi Raghunathan, Percy Liang, and Tengyu Ma. An explanation of in-context learning as implicit bayesian inference. arXiv preprint arXiv:2111.02080, 2021.
>
> [3] Lingyu Gao, Aditi Chaudhary, Krishna Srinivasan, Kazuma Hashimoto, Karthik Raman, and Michael Bendersky. Ambiguity-aware in-context learning with large language models, 2023.
>
> [4] Hanlei Zhang, Hua Xu, Ting-En Lin, and Rui Lyu. Discovering new intents with deep aligned clustering. In AAAI Conference on Artificial Intelligence, 2021b.
>
> [5] Rajat Kumar, Mayur Patidar, Vaibhav Varshney, Lovekesh Vig, and Gautam Shroff. Intent detection and discovery from user logs via deep semi-supervised contrastive clustering. In PConference of the North American Chapter of the Association for Computational Linguistics: Human Language Technologies, 2022.
>
> [6] Xiang Shen, Yinge Sun, Yao Zhang, and Mani Najmabadi. Semi-supervised intent discovery with contrastive learning. In Workshop on Natural Language Processing for Conversational AI, 2021.
>
> [7] Li, Haoran, et al. "MTOP: A comprehensive multilingual task-oriented semantic parsing benchmark." arXiv preprint arXiv:2008.09335 (2020).

---

### Official Review · Reviewer_G7hj · 2023-10-31

**Soundness:** 3 good
**Presentation:** 2 fair
**Contribution:** 3 good
**Rating:** 6
**Confidence:** 3

**Summary:**

The paper proposes a system of GPTs, called IntentGPT, in order to discover new intents in an open-world recognition system. The method of the paper consists of an LLM (or two) with a prompt to design an in-context learning prompt to do intent identification from utterances, with a special provision for labeling new intents that it has not seen before. This prompt is then augmented by including semantically relevant examples of labeled utterances to the test utterances and a list of possible intents and then handed to an LLM to perform the intent identification. The paper evaluates its method on two different benchmarks, with three different LLMs, and shows competitive, and often better, performance as compared to both semi-supervised and full unsupervised techniques for intent discovery.

**Strengths:**

The paper has good empirical validation is clear in its explanations and figures and is attacking a difficult and significant problem.  The paper does a thorough empirical validation by comparing the multiple LLM models, including an open-source, one. This highlights that the proposed methodology is actually what is working for the task at hand. The paper also shows its proposed methods' performance relative to state-of-the-art methods which thoroughly validates the methods' effectiveness. Overall, given the context and high levels of linguistics understanding required for such a task as open-world intent discovery, the use of LLMs is a solid approach.

The paper also proactively attempts to answer questions that arise when using a GPT model, such as whether the model has seen the training data before in its pre-training.

**Weaknesses:**

The paper is missing some grounding in previous literature and it's not clear what the value of all of the components of its proposed methodology is. First, the whole component of Semantic Few-shot Sampling (SFS) reads to me as just being Retrieval Augmented Generation or RAG. See https://ai.meta.com/blog/retrieval-augmented-generation-streamlining-the-creation-of-intelligent-natural-language-processing-models/ for a description. The SFS component then is not a novel contribution and should be cited as RAG.

Second, it's not clear from the write-up what the value of the in-context prompt generator (ICPG) step is. The authors mention  that they do an ablation study with this module but do not provide what the alternative prompt is besides “Utilizing the automatically generated In-Context Prompt (ICP) by IntentGPT proves superior to using a \textit{simple prompt description}.” From looking at the examples provided in the appendix, the crafted prompt looks like what I would have given the LLM as a base prompt for this task, so I am not convinced that the ICPG step is producing better prompts than what a human would give.

Third, and this is more of a minor criticism and possibly something for future work, but why not test other prompting schemes? For example, especially given the ICPG step, why not tree a Chain of Thought style of prompt for this task? Doing such a prompting scheme combined with the ICL might deliver even better results.

**Questions:**

There are a couple of questions about the manuscript, some of which have already been detailed earlier in this review.

-	What is the “simple prompt description” that was done in the ablation study of the ICPG module?
-	When doing empirical testing, since this is meant to be used in an open-world setting, was the inclusion of test examples that express no intent or multiple intents considered? If so, how did this method handle those cases?

---

> ### Author Response · Authors · 2023-11-23
>
> We thank the reviewer for reviewing our paper. We are encouraged to read that the reviewer finds our paper is attacking a difficult problem, and our methodology performs well compared to previous baselines. We now respond to their questions below.
>
> > The Semantic Few-shot Sampling (SFS) component reads to me as Retrieval Augmented Generation or RAG, then is not a novel contribution.
>
> The main difference is that Retrieval Augmented Generation (RAG) has been mainly explored in knowledge-intensive tasks like Question Answering [1], where factual information is retrieved to ground the generation, while we use it to extract few-shot examples, a much less explored problem [2, 3]. Therefore, we would rather consider the SFS and SKIF components as minor contributions to the paradigm of few-shot learning building upon the RAG model. We have updated our paper with this discussion, citing relevant work on RAG.
>
> [1] Lewis, Patrick, et al. "Retrieval-augmented generation for knowledge-intensive nlp tasks." Advances in Neural Information Processing Systems 33 (2020): 9459-9474.
>
> [2] Liu, Jiachang, et al. "What Makes Good In-Context Examples for GPT-$3 $?." arXiv preprint arXiv:2101.06804 (2021).
>
> [3] Izacard, Gautier, et al. "Few-shot learning with retrieval augmented language models." arXiv preprint arXiv:2208.03299 (2022).
>
> > Why not test other prompting schemes like Chain of Thought?
>
> Chain of Thought is an appealing research avenue in extending our research. Previous works that use Chain of Thought [1, 2], improve in-context learning capabilities by performing two forward passes: one for generating a rationale of the task and a second to perform the task using that rationale. We perform a similar approach by first using an LLM to generate an informed and domain-specific prompt, and then use it to perform the task with a second LLM. We coincide with the reviewer that exploiting this prompting scheme could further improve results. We have incorporated this direction for future work in the revised paper.
>
> [1] Wang, Xuezhi, et al. "Self-consistency improves chain of thought reasoning in language models." arXiv preprint arXiv:2203.11171 (2022).
>
> [2] Lee, Harrison, et al. "Rlaif: Scaling reinforcement learning from human feedback with ai feedback." arXiv preprint arXiv:2309.00267 (2023).
>
>  > I am not convinced that the ICPG step is producing better prompts than what a human would give.
>
> As seen in the generated prompts from the ICPG module (see Appendix A.2, Prompts 2 and 3), it creates new prompts with rich information about the dataset at hand, which is different from the “vanilla prompt” that a human can create (like the one seen in Appendix A.2. Prompt 1). As shown in responses to reviewers GqJc and NWU5 about the ablation studies depicted in Table 2,  the ICPG module is beneficial for obtaining performance boosts. Finally, let us show how the ICPG module creates better prompts than humans by testing it in a multi-lingual dataset, where a human might not know all the languages to manually create prompts. We have extended our experiments to the MTOP [1] dataset, and the results depict how IntentGPT's ICPG module generates a new prompt in the desired language automatically (see response to reviewer kRkh).
>
> [1] Li, Haoran, et al. "MTOP: A comprehensive multilingual task-oriented semantic parsing benchmark." arXiv preprint arXiv:2008.09335 (2020).
>
> > What is the “simple prompt description” that was done in the ablation study of the ICPG module?
>
> The simple prompt description is included in Appendix A.2 (Prompt 5: E. Basic human-generated task description) to act as a baseline for our ablation analysis.
>
> > Did you consider the inclusion of test examples that express no intent or multiple intents?
>
> In our experiments, we did not test instances that may not have an intent or have multiple intents associated. Nevertheless, to handle multiple intents, our method could be easily extended to handle multiple or no intents, by adjusting the original prompt description feed to the In-Context Prompt Generator and formatting the examples to support multiple intents in a single response. However, there are no public benchmarks for multiple intent or no intent, but we consider this as a very interesting task for future work.

---

### Official Review · Reviewer_NWU5 · 2023-11-01

**Soundness:** 2 fair
**Presentation:** 4 excellent
**Contribution:** 1 poor
**Rating:** 3
**Confidence:** 4

**Summary:**

This paper demonstrates the power of pre-trained LLM like GPT to both detect known dialogue intents and more importantly discover intents not previously known. The paper shows how the in-context prompt of an LLM can be crafted using examples from known intents, and how certain prompt design decisions affect the performance.

**Strengths:**

1.	This is a well-written paper, well-organized, and clear to read and follow.
2.	There is a substantive literature review which is well presented, clearly comparing each work to the paper.
3.	There are substantive appendices that are very helpful to the reader.
4.	Throughout all of it, there is obvious diligence, attention to detail, and care for the presentation.
5.	This work could be useful to anyone new to the area of Intent Discovery, and in particular applying LLMs to this problem.
a.	This is not in conflict with the low contribution score above since the contribution question relates to research innovation value and the ICLR audience expectations - see below for more on this point.

**Weaknesses:**

The first LLM does not appear to contribute any value to the model, as discussed below. If this criticism is correct, then this work becomes more of an application paper, a study of applying off-the-shelf LLM to a known problem, showing how readily available new technology outperforms previous methods while significantly increasing simplicity and flexibility. While valuable, it is probably not a good match for this conference/track.

The role of the first LLM (LLM1) in the model is to create the prompt for the second LLM (LLM2). LLM1 is given an original prompt created by a human and asked to craft an optimized prompt to be used with LLM2. This can be an effective method to optimize LLM performance, chaining LLMs like this. (The reviewer uses this technique quite a bit). However, LLM1 would usually use its intelligence to create something better, with a clear added value, compared to the original input. Often, LLM1 considers specialized data in this process that is later not observed by LLM2. None of this seems to be the case in the proposed model. The list below shows a complete generated prompt by LLM1 (from Appendix 2), and the original prompt given to LLM1 that was the basis for its generation. It is obvious that the generated prompt is at most a slight rewording of the original. Next, the auxiliary data (few shot examples, known intents) given to LLM1 are basically just passed to LLM2, i.e. they could just as well be given directly to it. In other words, nothing suggests that the performance would not be equivalent if the model just used one LLM, with the original prompt and the auxiliary data (few shot examples, known intents) added to it. At the very least, the paper should compare to this scenario. The ablation study result for ICP, if I understand it correctly, compares to using a “simple prompt” like Prompt 5.E. in Appendix 2 – which lacks the key addition of the auxiliary data (which are very informative to an LLM like GPT), and is therefore not a true comparison of using one vs two LLMs.

LLM1 Generated Prompt: “As an AI language model, your task is to assign the correct intent to a given textual utterance.”
Original Prompt: “You are a helpful assistant and …  You specialize in … the task of assigning textual utterances to specific intents”

LLM1 Generated Prompt: “The intent can be one of the pre-defined intents or a new one that you create based on the context and knowledge about the problem and specific data domain.”
Original Prompt: “… some of which are pre-defined and others are not and have to be created”… “sufficient context and knowledge about the problem and specific data domain.”

LLM1 Generated Prompt: “You should never assign an utterance to ’unknown’.”
Original Prompt: “never assign a utterance to ’unknown’”

LLM1 Generated Prompt: “For each utterance, analyze the context and the specific request or action implied.”
Original Prompt: “…acquire sufficient context…”

LLM1 Generated Prompt: “If the utterance matches a known intent, assign it to that intent. If it doesn’t match any known intent, create a new intent that accurately represents the request or action implied by the utterance.”
Original Prompt: “you need to be aware of the known intents and reuse them as much as possible, but need to create new intents when there are not known intents that fit the given utterance”

LLM1 Generated Prompt: “Remember, the goal is to understand the user’s intent as accurately as possible.”
Original Prompt: “maximizing the model’s performance in the task”

LLM1 Generated Prompt: “Be aware of the known intents and reuse them as much as possible, but don’t hesitate to create new intents when necessary.”
Original Prompt: “be aware of the known intents and reuse them as much as possible, but need to create new intents”

The other generated prompt example in Appendix 2 (Prompt 3.C) is basically equal to the original as above, with the exception of the following fragment: “The utterances can be questions, statements, or requests related to banking services like card transactions, account top-ups, refunds, identity verification, card delivery, transfer fees, and more.” – But this is just a summary of the auxiliary example data and a single GPT would have it by default by having the auxiliary example data itself.

A couple of smaller points:
1.	For the “Semantic Few-Shot Sampling” the proposed model uses SentenceBERT to do the embeddings, which is a strange choice given that better quality embeddings can be attained with OpenAI’s GPT3+ models, which are already used in this work in other ways.
2.	On page 8, “that while SKIF” should probably be “that SKIF”

**Questions:**

Would you be able to run a comparison to a one-LLM setup as described above, i.e. by using the “Original Prompt” and adding the auxiliary data directly to it?

---

> ### Author Response · Authors · 2023-11-23
>
> We thank the reviewer for taking the time to review our paper very carefully. We are pleased to find that the reviewer feels the presentation of the paper and literature review were quite good, and the substantial appendices included are helpful for the reader. We now address their concerns regarding the contribution of our work.
>
> > Does the first LLM, In-Context Prompt (ICP), contribute to any value to the model?
>
> Yes, the first LLM, In-Context Prompt Generator (ICPG), adds value to our model. The setup described by the reviewer is present in the paper. Please see our ablation studies, detailed in Table 2, which shows a comparison between the use of original prompts with auxiliary data (a single LLM with few-shot (FS) and Known Intent Feedback (KIF)) against our ICP-enhanced prompts.
>
> Specifically, Rows 2, 3, and 4 of Table 2 of the paper (for both GPT3.5 and GPT4 versions) show the baseline performance using a single LLM, while Rows 5 and 6 highlight the improvements with the ICP module activated. The results show a consistent performance gain of 1-2 points across both datasets and various GPT models, underscoring the efficacy of our approach.
>
> Additionally, thanks to a new experiment run on a multi-language dataset suggested by reviewer kRkh, we can show how the first LLM is able to handle the language adaptation without human effort, only relying on a few context examples from the train set. We find this case relevant for highlighting the contribution of ICP.
>
> > The choice of SentenceBERT to compute embeddings is a strange choice. Did you try OpenAI’s GPT3+ models?
>
> We tested text-embedding-ada-002 embeddings and we observed that they perform worse than SentenceBert (see table below). The most likely reason for OpenAI embeddings being worse is that it is primarily used to encode large contexts like documents for document retrieval, whereas SentenceBert is trained to efficiently encode short contexts, or sentences, like the size of our intents. We have incorporated this result in the Appendix of the paper.
>
> | Model                | Embeddings | KIR | NMI   | ACC   | ARI   |
> |----------------------|-----|-----|-------|-------|-------|
> |               |                    **CLINC dataset**     |           |       |       |       |
> | GPT-3.5 (10 shot)     |  text-embedding-ada-002   | 0.75| 88.64| 71.36| 61.42 |
> | GPT-3.5 (10 shot)     |   SentenceBert   | 0.75| 91.66| 78.99| 71.58 |
> | GPT-4 (10 shot)     |  text-embedding-ada-002   | 0.75| 90.73| 74.89| 65.99 |
> | GPT-4 (10 shot)     |   SentenceBert   | 0.75| 94.01| 83.96| 77.17 |
> |              |                   **BANKING dataset**     |     |     |     |       |       |       |
> | GPT-3.5 (10 shot)     |   text-embedding-ada-002  | 0.75| 69.46| 52.12| 39.51 |
> | GPT-3.5 (10 shot)      |  SentenceBert   |  0.75| 88.21 | 82.42 | 88.74 |
> | GPT-4 (10 shot)     |   text-embedding-ada-002  | 0.75| 88.64| 71.36| 61.42 |
> | GPT-4 (10 shot)      |  SentenceBert   |  0.75| 91.66| 78.99| 71.58 |

---

### Official Review · Reviewer_GqJc · 2023-11-03

**Soundness:** 4 excellent
**Presentation:** 3 good
**Contribution:** 2 fair
**Rating:** 5
**Confidence:** 4

**Summary:**

The paper studies the problem of intent classification/clustering and discovery using in-context learning and LLMs. It proposes 1) A method to discover useful prompts for the problem using an LLM 2) Use those prompts with few shot examples (based on semantic similarity) to classify utterances and discover new intents. The latter step is executed iteratively which leads to a growing database of novel intents.

**Strengths:**

1. The paper is written well and cites the relevant literature
2. Intent classification and discovery is an interesting problem on its own
3. It also has many potential useful downstream applications
4. The results improve upon prior baselines

**Weaknesses:**

While studying LLMs and their performance is an important endeavour, I am not convinced by the novelty of the proposed method or the efficacy of the individual components. For instance,

1. The first part of the pipeline (ICP) seems to add little value. From row 3 vs row 5 in bottom panel of Table 2, it seems that on average ICP led to no performance boost.
2. The semantic few shot sampling (SFS) also seems to add limited value (row 3 vs row 4)
3. Intutitively "Feedback" should help, but I am not sure how is performance affected (see question below).
4. Is row 1 (i.e. no tickmarks) of Table 2 (top and bottom panels) the right baseline to compare with? Based on the appendix it seems that for row 1, the KIR ratio is 0 (since the prompt contains no information about the space of labels) while the remaining rows presumably have a KIR of 0.75.
5. How does 50-shot vanilla GPT4 with KIR = 0.75 perform? Is that closer to row 3 of table 2 or row 0?

As mentioned in the appendix, GPT4 was trained on an unknown data distribution. The authors point out, that it clearly knows about the dataset. The arguments presented in the appendix about GPT4 not having seen the test/train split are unconvincing. I am not sure what a Frechet distance of 0.54 means without a control, but Table 5 seems to suggest that for 5 out of 12 categories, the predicted intent exactly matches the ground intent. Is that evidence that GPT4 knows a lot more than just the name of the datasets?

Is the fact the the optimal clusters (155 and 88) are so close to the ground truth numbers (150 and 70) an indication of dataset leakage or is it just an inevitable side effect of the evaluation criteria?

**Questions:**

1. How does performance change if only the feedback part is ablated (i.e. the database of intents is constant) from the entire pipeline?
2. It would be nice to have a longer version on Appendix A.7 with more examples and details about the statistics of discovered intents. How close are they semantically/syntactically to the ground truth labels?

---

> ### Author Response · Authors · 2023-11-23
>
> We thank the reviewer for their time spent reviewing the paper. We are encouraged to read that you see our method has useful downstream application potential and outperforms existing baselines. We now respond to their comments in detail below.
>
> > The In-Context Prompt (ICP) seems to add little value. From row 3 vs row 5 in the bottom panel of Table 2, it seems that on average ICP led to no performance boost.
>
> While the difference between rows 3 vs row 5 does not lead to a significant average performance boost, the scores between rows 4 and row 6 when adding ICP (both using SFS) indicate that it boosts performance. Notably, comparing rows 2, 3, and 4 (which do not use ICP) with rows 5 and 6 (which use the ICP), as a whole, shows a consistent improvement between 1-2 points in all the settings (across datasets and GPT models), except for the case of ARI on CLINC dataset for GPT3.5, where results remain similar (see also responses to reviewers NWU5 and kRkh on this matter). As this is an ablation study, a component may not boost performance in all scenarios but when combined with the other parts of our pipeline we show that ICP does help. ICP aims to redirect the prompt towards a specific domain without human labor, offering a boost in performance as seen by our ablations.
>
> > Is row 1 of Table 2  the right baseline to compare with? Iit seems that for row 1, the KIR ratio is 0 while the remaining rows have a KIR of 0.75.
>
> As noted by the reviewer, row 1 in the ablation studies from Table 2 displays the zero-shot setting, which implies a KIR of 0, while others use a KIR of 0.75. While this comparison may be not well-suited for judging performance on the main metrics (i.e. NMI, ACC, ARI) it is interesting to observe how the Number of Discovered Intents (NDI) explodes. This is because this simple setting performs as a zero-shot, and there’s no regularization of the label space introduced by the Known Intent Feedback technique. Additionally, this setting shows how other metrics downgrade because of it. We have updated the paper by specifying that this row shows a zero-shot setting, and clarifying that a more fair comparison in terms of Known Intent Ratio can be done starting from rows 2-7.
>
> > How does 50-shot vanilla GPT4 with KIR = 0.75 perform? Is that closer to row 3 of table 2 or row 0?
>
> A similar question was asked by reviewer NWU5, where they asked for 10-shot vanilla GPT4 with KIR = 0.75. This ablation is shown in rows 2, 3, and 4 of Table 2 in the paper, which presents the performance boosts of including the ICP module when comparing against rows 4 and 6 (Please see our comment to reviewer NWU5 highlighting the results). Further, to answer your question, we also ran an experiment with 50-shot vanilla GPT4 and KIR=0.75. We present the results as follows:
>
>
> | Model                 | KIR | NMI   | ACC   | ARI   |
> |----------------------|-----|-------|-------|-------|
> |               |                    **CLINC dataset**     |           |       |       |       |
> | GPT-3.5 (50 shot)       | 0.75| 90.38|75.11|67.43 |
> | GPT-4 (50 shot)       | 0.75| 92.59|81.38|71.43 |
> |              |                   **BANKING dataset**     |     |     |     |       |       |       |
> | GPT-3.5 (50 shot)     |   0.75| 77.29|60.85|45.9  |
> | GPT-4 (50 shot)     |    0.75| 81.72|67.76|55.7 |
>
> Comparing this result with Table 2, this setting would perform similarly to rows 2 and 3, and far from the top positions. Even though this setting uses a lot of few-shot samples, it is not able to get better results because of the vanilla prompt. Note that when adding the ICP to the 50-shot setting, we obtain our best results, with an NMI=96.06 on CLINC, showing the contribution of having the ICPG component.

---

> ### Author Response · Authors · 2023-11-23
>
> > How does performance change if only the feedback part is ablated (i.e. the database of intents is constant) from the entire pipeline?
>
> We have run this experiment and present the results below.
>
> | Model                 | KIR | NMI   | ACC   | ARI   |
> |----------------------|-----|-------|-------|-------|
> |               |                    **CLINC dataset**     |           |       |       |       |
> | GPT-3.5 (10 shot)       | 0.75| 87.62|57.71|54.77 |
> | GPT-4 (10 shot)       | 0.75| 87.76|59.69|56.57 |
> |              |                   **BANKING dataset**     |     |     |     |       |       |       |
> | GPT-3.5 (10 shot)     |   0.75| 77.98|40.99|35.87  |
> | GPT-4 (10 shot)     |    0.75| 76.54|37.27|32.66 |
>
> Here, we study the impact of maintaining a constant intent database, only composed of the known intents at initialization, and not growing the database during the evaluation. We observe a degradation of the performance in all metrics, compared to rows 2-7 in the ablation table (consider these rows also in GPT-4). However, this setting outperforms the baseline shown in row 1 of the ablation, as we include the mechanism of injecting Intents into the prompt. We also display NDI, which explodes due to the lack of intent reuse. We have added the findings from this additional experiment to our ablation study in the paper.
>
> > The optimal clusters (155 and 88) are so close to the ground truth (150 and 70). Is this an indication of dataset leakage?
>
> We do not use GPT-4 or a trained model to determine the optimal number of clusters. We utilize DBSCAN on the predicted intent embeddings to determine the optimal K for k-means clustering. Therefore, there can not be dataset leakage.
>
> > Sometimes the predicted intents match exactly the ground intent. Is that evidence that GPT4 knows a lot more than just the name of the datasets?
>
> We thank the reviewer for taking the time to think about this problem in our Intent Discovery setup. We find that this concern is valid. Therefore, we propose the Frechet Bert Distance as a reasonable measure to compute the similarity between the predicted intent and the ground truth distributions to identify whether the model is regurgitating its predictions from memorization. We see that there is an acceptable level of dissimilarity between the real and predicted intents, with a score of 0.54. A perfect match would be 0 and a complete dissimilarity would be 1. As shown in the paper, we are not able to reproduce utterances or labels from the test sets by manually interacting with the model.
>
> Notably, the problem of detecting pre-train data in LLMs has recently received attraction [1, 2], however, there is not yet a standard approach on how the decide that a text example was part of the pre-training data. We will explore this direction in future work.
>
> [1] Shi, Weijia, et al. "Detecting Pretraining Data from Large Language Models." arXiv preprint arXiv:2310.16789 (2023).
> [2] Meeus, Matthieu, et al. "Did the Neurons Read your Book? Document-level Membership Inference for Large Language Models." arXiv preprint arXiv:2310.15007 (2023).
>
> > It would be nice to have a longer version of Appendix A.7 with more examples and details about the statistics of discovered intents. How close are they semantically/syntactically to the ground truth labels?
>
> We have included a complete list of results in the supplementary material (see results.csv). We also extended Appendix 7 with a study on the complete sets of discovered intents and ground truth intents. In that study, we compute SentenceBert embeddings from the two sets and visualize them using TSNE in a 2D grid. We observe that some intents coincide, and many others do not. The ones that coincide correspond to samples with a straightforward intent naming choice, while the others require more naming creativity. Note that the required capability for a good performance is to use intents robustly, assigning the same intent to utterances with the same ground truth intent, regardless of the exact name.
>
> In the extended list of discovered intents, we also observe that there is a reasonable difference between the predicted intents and the ground truth despite their semantic similarity. We consider that IntentGPT-3.5 and IntentGPT-4, conditioned with the proposed prompt scheme, offer sufficient generalization capability to not solve the task by memorization.

---

### Author Response · Authors · 2023-11-23
**General comment**

We thank the reviewers for their thoughtful feedback and careful evaluation of our paper. We are encouraged by the overall positive tone of the reviews, recognizing our work's soundness and the presentation's quality.

The reviewers have noted that our work *“could be useful to anyone new to the area of Intent Discovery”* and that it *“has many potential useful downstream applications”* (NWU5). Further, they observe that *“the paper has good empirical validation is clear in its explanations [...] and is attacking a difficult and significant problem”* and that *“given the [...] high levels of linguistics understanding required for such a task [...] the use of LLMs is a solid approach”* (G7hj). Reviewers recognize that our method *“eliminates the need for training and extensive domain-specific data”* (kRkh) and that our work offers *“good prompt engineering to guide GPT-4 to produce intents”* (vjHF). Finally, reviewers KRkh, vjHF, G7hj, and GqJc coincide in acknowledging that IntentGPT outperforms previous baselines on Intent Discovery benchmarks.

We highly appreciate the constructive feedback and we have revised the paper accordingly. We follow by addressing reviewers' concerns separately, providing clarifications about our approach and results.

---

### Meta-Review · Area_Chair_uGqY · 2024-01-06

**Metareview:**

The authors propose an LLM-based method for intent discovery (i.e., open-domain classification) using a two stage process of: (1) in-context prompt generation that includes markers for content-specific injection and (2) an intent predictor which instantiates the generated prompt with specific content derived from designed procedures (e.g., few-shot examples from a few-shot sampler, known intent feedback using most similar existing intents, test examples). [see Figure 2 for a very easily understandable example; this is referred to as 'IntentGPT']. This procedure can either classify the utterance into one of the existing intents or create a new intent (and add it to the set of known intents). IntentGPT is empirically evaluated on two widely used intent classification/discovery datasets in the initial submission and SNIPS, StackOverflow, and MTOP (which is multi-lingual) during rebuttal, showing SotA performance in most cases relative to several recent unsupervised/ZSL and semi-supervised/FSL baselines with widely used metrics for this task (and with multiple LLM candidates).

Consensus strengths identified by reviewers regarding this submission include:
- The paper is well-written, easy to understand, and is generally well-contextualized with respect to existing relevant literature. In particular, the paper can be largely understood by just reading the abstract+figures+caption and the appendices help elucidate details.
- Intent discovery is an increasingly more studied problem than (closed-domain) intent discovery.
- The empirical results are exhaustive (especially after rebuttal) and demonstrates convincing improvements.

Conversely, consensus (or interesting individual) concerns/limitations included:
- There were concerns regarding in-context prompt generation (which is conceptually understandable as it seems this is likely relatively static). This was addressed in rebuttal reasonably well and more convincingly once you consider the MTOP data setting. However, the reviewers weren't entirely convinced (and I am on the fence).
- This work is closely tied to the application as it isn't clear that it adds anything conceptually that would transfer to other problems. I believe this criticism is also relevant to LLMs in general, where intent classification/discovery is more for validation than having clear independent utility (except in NLP settings where this is of academic interest).
- Instead of a two-stage prompting procedure, couldn't we use a chain-of-thought like model. While this could be left for future work, it is a fairly obvious question in my opinion.

In my opinion, the initial submission was a very targeted application for a historically important task (that may be of decreasing importance as-is) with a straightforward, but well-executed method (and notably well-written paper). The additional empirical results added during rebuttal make for a much more interesting paper that opens up additional avenues for investigation and potential to influence other works. However, these new results haven't been integrated into the paper and thus it doesn't have sufficient discussion, etc. In combination with concerns regarding a two-stage process as opposed to a multi-stage LLM (e.g., chain of thought), it still isn't clear that the results extend beyond intent discovery. Therefore, I think it is an interesting paper, but the impact will be limited to this task.

**Justification For Why Not Higher Score:**

This paper is focused on a particular application and the findings aren't likely to influence methods beyond this application. Additionally, there are newer LLM prompting methods that weren't considered and would be applicable. Finally, the additional evidence provided during rebuttal need more discussion as it is more than twice the set of original experiments.

**Justification For Why Not Lower Score:**

N/A

---

### Decision · Program_Chairs · 2024-01-16

Reject